# HRI coordinates translation necessary for protein homeostasis and mitochondrial function in erythropoiesis

Shuping Zhang[1], Alejandra Macias-Garcia[1], Jacob C Ulirsch[2,3,4,5], Jason Velazquez[1], Vincent L Butty[6], Stuart S Levine[6], Vijay G Sankaran[2,3,4], Jane-Jane Chen[1]*

[1]Institute for Medical Engineering and Science, Massachusetts Institute of Technology, Cambridge, United States; [2]Division of Hematology/Oncology, Boston Children's Hospital, Harvard Medical School, Boston, United States; [3]Department of Pediatric Oncology, Dana-Farber Cancer Institute, Harvard Medical School, Boston, United States; [4]Broad Institute of Massachusetts Institute of Technology and Harvard, Cambridge, United States; [5]Program in Biological and Biomedical Sciences, Harvard University, Cambridge, United States; [6]BioMicro Center, Massachusetts Institute of Technology, Cambridge, United States

**Abstract** Iron and heme play central roles in the production of red blood cells, but the underlying mechanisms remain incompletely understood. Heme-regulated eIF2α kinase (HRI) controls translation by phosphorylating eIF2α. Here, we investigate the global impact of iron, heme, and HRI on protein translation in vivo in murine primary erythroblasts using ribosome profiling. We validate the known role of HRI-mediated translational stimulation of integratedstressresponse mRNAs during iron deficiency in vivo. Moreover, we find that the translation of mRNAs encoding cytosolic and mitochondrial ribosomal proteins is substantially repressed by HRI during iron deficiency, causing a decrease in cytosolic and mitochondrial protein synthesis. The absence of HRI during iron deficiency elicits a prominent cytoplasmic unfolded protein response and impairs mitochondrial respiration. Importantly, ATF4 target genes are activated during iron deficiency to maintain mitochondrial function and to enable erythroid differentiation. We further identify GRB10 as a previously unappreciated regulator of terminal erythropoiesis.
DOI: https://doi.org/10.7554/eLife.46976.001

*For correspondence:
j-jchen@mit.edu

Competing interests: The authors declare that no competing interests exist.

## Introduction

Iron-deficiency anemia is estimated to affect one-third of the global population (*Camaschella, 2019*). Iron and heme are key components of hemoglobin, the primary oxygen transport molecule, and their cellular levels impact globin synthesis and red blood cell production. Specifically, globin is transcriptionally regulated by BACH1 (BTB Domain and CNC homolog 1) (*Igarashi and Sun, 2006*) and is regulated at the level of protein translation by HRI (heme-regulated eIF2α kinase) (*Chen, 2007*), both of which are heme-sensing proteins.

During heme deficiency that is induced by dietary iron deficiency (ID) in mice, HRI is activated and phosphorylates the α subunit of eukaryotic initiation factor eIF2 (eIF2α) to inhibit translation of α- and β-globin mRNAs, thereby preventing proteotoxicity resulting from heme-free globin chains (*Han et al., 2001*). Meanwhile, phosphorylated eIF2α (eIF2αP) selectively enhances the translation of activating transcription factor 4 (*Atf4*) mRNA (*Suragani et al., 2012*; *Chen, 2014*). This coordinated translational repression of general protein synthesis with the specific translational enhancement of

**eLife digest** Red blood cells use a molecule called hemoglobin to transport oxygen around the body. To make hemoglobin, cells require iron to build a component called heme. If an individual does not get enough iron in their diet, the body cannot produce enough red blood cells, or the cells lack hemoglobin. This condition is known as iron deficiency anemia, and it affects around one-third of the world's population.

Researchers did not know exactly how iron levels control red blood cell production, though several proteins had been identified to play important roles. Heme forms in the cell's mitochondria: the compartments in the cell that supply it with energy. When heme levels in a developing red blood cell are low, a protein called HRI reduces the production of many proteins, most importantly the proteins that make up hemoglobin. HRI also boosts the production of a protein called ATF4, which switches on a set of genes that help both the cell and its mitochondria to adapt to the lack of heme. In turn, HRI and ATF4 reduce the activity of a signaling pathway called mTORC1, which controls the production of proteins that help cells to grow and divide.

To understand in more detail how iron and heme regulate the production of new red blood cells, Zhang et al. looked at immature red blood cells from the livers of developing mice. Some of the mice lacked the gene that produces HRI, and some experienced iron deficiency. Comparing gene activity in the different mice revealed that in the developing blood cells of iron-deficient mice, HRI largely reduces the rate of protein production in both the mitochondria and the wider cell. At the same time, the increased activity of ATF4 allows the mitochondria to carry on releasing energy and the cells to continue developing. Zhang et al. also found that a protein that inhibits the mTORC1 signaling pathway needs to be active for the new red blood cells to mature.

Overall, the results suggest that drugs that activate HRI or block the activity of the mTORC1 pathway could help to treat anemia. The next step is to test the effects that such drugs have in anemic mice and cells from anemic people.

DOI: https://doi.org/10.7554/eLife.46976.002

*Atf4* mRNA by eIF2αP is termed the integrated stress response (ISR) (*Harding et al., 2003*). The ISR is a universal response to several types of cellular stress (*Chen, 2014*; *Pakos-Zebrucka et al., 2016*) and it is initiated by members of the eIF2α kinase family. Besides HRI, mammalian cells have three additional eIF2α kinases, which are expressed in distinct tissues and which combat specific physiological stress. PKR responds to viral infection (*Kaufman, 2000*), GCN2 senses nutrient starvations (*Hinnebusch, 1996*), and PERK is activated by endoplasmic reticulum (ER) stress (*Ron and Harding, 2000*). All four eIF2α kinases respond to oxidative and environmental stresses.

In the erythroid lineage, HRI expression increases during differentiation, with higher expression occurring in hemoglobinized erythroblasts (*Liu et al., 2008a*). Starting at the basophilic erythroblast stage, HRI is the predominant eIF2α kinase and is expressed at levels that are two orders of magnitude greater than those of the other three eIF2α kinases (*Kingsley et al., 2013*). At these stages, HRI is responsible for over 90% of eIF2α phosphorylation (*Liu et al., 2008b*). HRI-ISR is necessary for effective erythropoiesis during ID and acts by reducing oxidative stress and promoting erythroid differentiation (*Suragani et al., 2012*; *Zhang et al., 2018*). Furthermore, HRI-ISR represses the mTORC1 signaling that is activated by the elevated erythropoietin (Epo) levels occurring during ID specifically in the erythroid lineage (*Zhang et al., 2018*). Thus, HRI coordinates two key translation-regulation pathways, eIF2αP and mTORC1, during ID. However, the exact molecular mechanisms through which iron and heme regulate erythropoiesis are incompletely understood.

Mitochondria not only are the energy powerhouses of the cell, but also are necessary for amino acid metabolism, nucleotide production, and the biosynthesis of heme and iron-sulfur clusters (*Shpilka and Haynes, 2018*). Translational regulation of mitochondrial biogenesis by mTORC1 is particularly important for erythropoiesis because of the high demand of heme for hemoglobin production and oxidative stress (*Liu et al., 2017*). However, the roles of HRI and eIF2αP in mitochondrial biogenesis and function are still unknown.

Transcriptional regulation during erythropoiesis has been studied extensively (*Kerenyi and Orkin, 2010*; *An et al., 2014*), but much less is known about the translational control of this process

(*Mills et al., 2016*; *Khajuria et al., 2018*). Ribosome profiling (Ribo-seq) has emerged as a powerful tool that can be used to interrogate translation genome-wide (*Ingolia et al., 2009*). Here, we performed Ribo-seq and mRNA-seq in primary basophilic erythroblasts to investigate how in vivo translation is regulated by iron, heme, and HRI in order to gain a global understanding of the molecular mechanisms that govern erythropoiesis. We hypothesized that by globally surveying the landscape of translational and concomitant transcriptional changes that occur in the context of HRI deficiency (comparing the changes seen in iron replete (+Fe) or iron deficiency (–Fe) conditions), we could gain important insights into the mechanisms through which iron and heme regulate the process of erythropoiesis. Our results demonstrate that heme and HRI mediate the translation of both cytosolic and mitochondrial ribosomal protein mRNAs. Furthermore, HRI–ATF4 mediated gene expression is essential during ID to prevent the accumulation of unfolded proteins in the cytoplasm, to maintain mitochondrial oxidative phosphorylation, and to enable erythroid differentiation in developing basophilic erythroblasts.

## Results

### Overview of Ribo-seq and mRNA-seq data

Beginning at the basophilic erythroblast stage, erythropoiesis is finely regulated by iron and heme levels (*Chiabrando et al., 2014*; *Muckenthaler et al., 2017*). Thus, basophilic erythroblasts (hereafter referred to as EBs for simplicity) from Wt +Fe, Wt –Fe, Hri$^{-/-}$ +Fe, and Hri$^{-/-}$ –Fe fetal livers (FLs) were used as sources to generate Ribo-seq and mRNA-seq libraries for genome-wide analysis of transcriptional and translational changes (*Figure 1A*).

After applying standard protocols of quality control and preprocessing to remove rRNA and tRNA, we obtained 9.7–41.1 (median 26.2) million reads of Ribo-seq and 29.4–66.6 (median 42.5) million reads of mRNA-seq for subsequent mappings (*Supplementary file 1a*). As expected, most of the reads from both Ribo-seq (84–88%) and mRNA-seq (72–74%) were mapped to protein coding sequence (CDS), whereas some reads mapped to the 5′ and 3′ UTR or to other regions, mostly introns and the regions around transcription start sites (*Figure 1B*).

Our Ribo-seq data displayed excellent triplet periodicity, CDS enrichment, and limited 3′ UTR reads, validating the high quality of the ribosome-protected fragments (RPFs) that we obtained (*Figure 1B–C*). The proportions of reads mapping to the 5′ UTRs of Ribo-seq data were similar to those from mRNA-seq (*Figure 1B*), in agreement with reports of pervasive translation outside of annotated CDSs (*Ingolia, 2014*). Overall, 62.9% of the expressed genes in mRNA-seq were detected in Ribo-seq, and therefore appeared to be actively translated in EBs (*Figure 1D*). As shown in *Figure 1D*, 99% of the mRNAs detected in Ribo-seq data (with reads greater than 25 in at least one of the conditions) were present in the mRNA-seq data, demonstrating the high quality of our Ribo-seq data and the correlation of RPFs and mRNAs (*Figure 1E*).

### Upregulation of the in vivo translation of ISR mRNAs in WT EBs compared to HRI$^{-/-}$ EBs

A total of 317 mRNAs were significantly differentially translated between Wt and Hri$^{-/-}$ EBs in the +Fe condition (*Figure 2A* and *Supplementary file 1b*), supporting the role of HRI during normal fetal erythropoiesis. Twenty-five differentially translated mRNAs were common under both +Fe and –Fe conditions; these included the well-characterized ISR mRNAs, *Atf4*, *Ppp1r15a*, and *Ddit3* (*Figure 2A–B* and *Figure 2—figure supplement 1A*) as illustrated in Figure 8. *Atf4* mRNA was the most differentially translated mRNA between Wt and Hri$^{-/-}$ cells during ID (8.8-fold increase in translational efficiency (TE)), followed by the *Ddit3* (8.1-fold) and *Ppp1r15a* (3.7-fold) mRNAs (*Figure 2B* and *Supplementary file 1b*). Each of these mRNAs contains upstream open reading frames (uORFs) in their 5′ UTR, the use of which is upregulated by eIF2αP in cell lines under endoplasmic reticulum (ER) tostress or amino-acid starvation (*Pavitt and Ron, 2012*). We observed HRI-dependent regulation of the translation of these mRNAs involving ribosome occupancies in uORFs in vivo (*Figure 2C–D* and *Figure 2—figure supplement 1B–C*). We also verified that changes in TE corresponded to concordant changes in ATF4 protein levels in EBs (*Figure 2E*).

As *Eif2ak1* (HRI) and *Atf4* are among the most highly expressed and efficiently translated mRNAs in Wt EBs (top 3%, *Supplementary file 1c*), we investigated whether the translation of *Atf4* mRNA

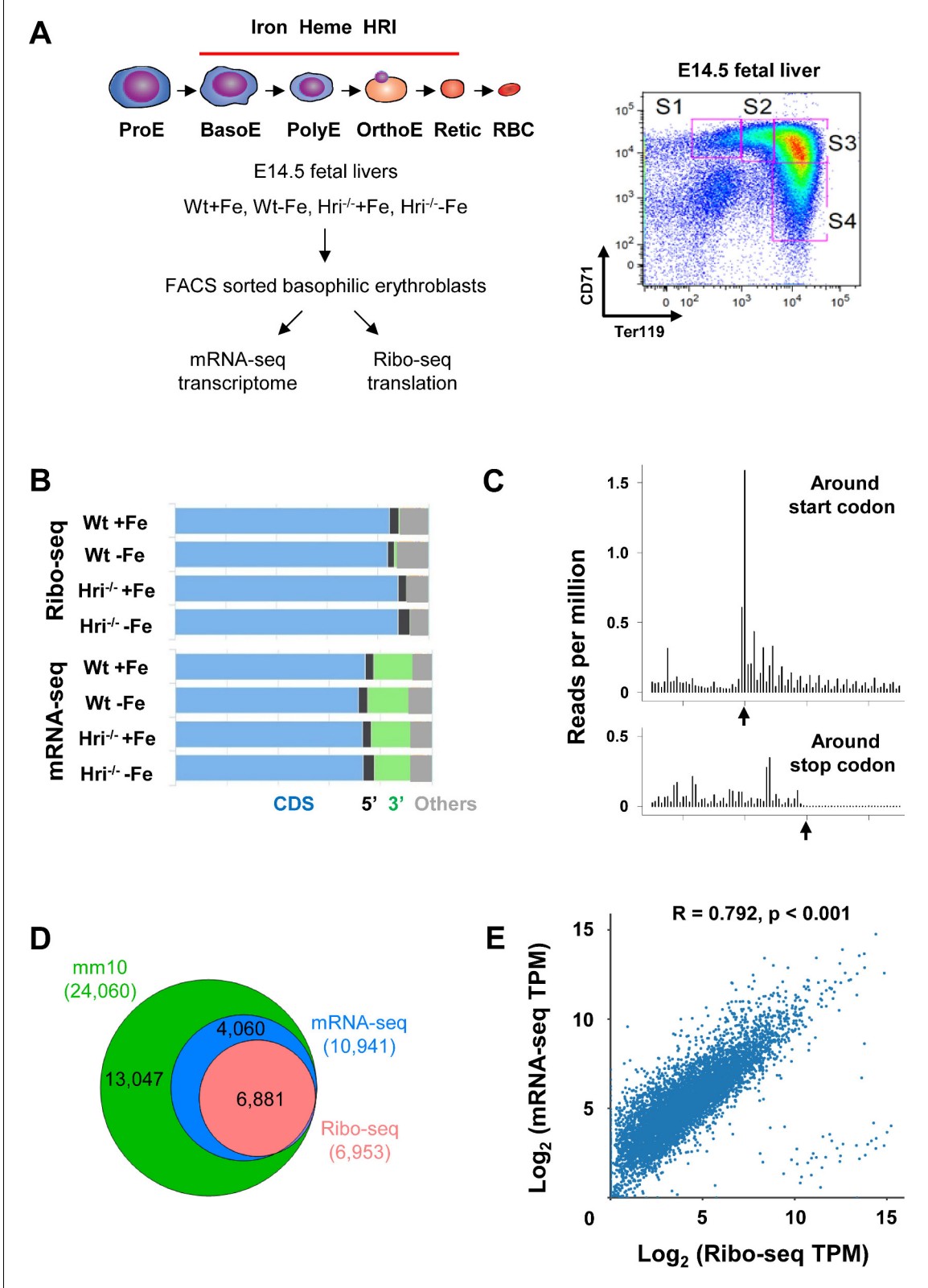

**Figure 1.** Overview of Ribo-seq and mRNA-seq data. (**A**) Illustration of experimental designs. Basophilic erythroblasts (EBs) (S3) from E14.5 FLs of Wt +Fe, Wt –Fe, Hri$^{-/-}$ +Fe and Hri$^{-/-}$ –Fe embryos were sorted and subjected to Ribo-seq and mRNA-seq library preparations. (**B**) Distribution of the mapped reads from Ribo-seq and mRNA-seq from one replica. (**C**) A representative plot of the triplet periodicity of Ribo-seq from Wt –Fe EBs. Arrows indicate the start and stop codons. (**D**) Gene coverages of Ribo-seq and mRNA-seq data in the mouse genome (UCSC, mm10). (**E**) Scatter plot and

*Figure 1 continued on next page*

*Figure 1 continued*

correlation analysis of log$_2$-transformed TPM (transcript per million) of Ribo-seq and mRNA-seq data from Wt +Fe EBs. E, Erythroblast; RBC, red blood cell; Retic, Reticulocyte.

DOI: https://doi.org/10.7554/eLife.46976.003

is regulated by HRI in vivo. Atf4 mRNA has two well-characterized uORFs in its 5′ UTR. uORF1 encodes three amino acids and is translated regardless of stress and eIF2αP levels (*Pavitt and Ron, 2012*). We observed that uORF1 had exceptionally high ribosome occupancy (5.7-fold and 21.3-fold higher than *eIF2s1* (eIF2α) and *Rps6* initiating AUGs, respectively (*Supplementary file 1d*)). However, uORF2 and the canonical ORF of *Atf4* were poorly translated in Hri$^{-/-}$ compared to Wt EBs under both +Fe and −Fe conditions (*Figure 2C–D*). Together, these data support the idea that *Atf4* mRNA is primed for translation by HRI in developing EBs, and that HRI is a master translational regulator of key ISR mRNAs in vivo in primary EBs, especially during ID.

## Upregulation of translation of cytosolic and mitochondrial ribosomal proteins in HRI$^{-/-}$ −Fe EBs compared to Wt −Fe EBs

Gene Ontology (GO) analysis of the differentially translated mRNAs revealed that the translation of components of ribosomes and mitochondria is most significantly upregulated in HRI- and iron-deficient states (*Figure 3A*). Gene Sets Enrichment Analysis (GSEA) of differentially translated mRNAs further showed that translation of mRNAs that are involved in ribosome synthesis, mTORC1 signaling, and translation initiation and elongation were most highly elevated in Hri$^{-/-}$ −Fe EBs compared to Wt −Fe EBs (*Figure 3B–C*). Increased 43S translation initiation complex formation in Hri$^{-/-}$ −Fe cells (*Figure 3B*) is consistent with the known function of HRI–eIF2αP in ternary and 43S complexes formation (*Chen, 2007*). Furthermore, we observed that *Eef1a1* and *Ccnd3*, which have been shown to be regulated by mTORC1 signaling (*Thoreen et al., 2012*), were also more efficiently translated in Hri$^{-/-}$ −Fe EBs (*Figure 2B*). Notably, 56 ribosomal protein (RP) mRNAs were more highly translated in Hri$^{-/-}$ −Fe EBs than in Wt-Fe EBs (*Figure 3D*, *Figure 3—figure supplement 1A–B* and *Supplementary file 2a*). Whereas 38 of these RP mRNAs were known to be regulated by mTORC1 via 5′ terminal oligopyrimidine (TOP) motifs in their 5′ UTR (*Thoreen et al., 2012*), translation of the other 18 RP mRNAs was regulated by HRI, but not mTORC1. There was less overlap for non-RP targets between mTORC1- and HRI-mediated translation. Among the 20 non-RP mRNAs, translation of 12 mRNA was regulated by HRI but not by mTORC1 (*Figure 3D* and *Supplementary file 2a*).

The second set of most preferentially translated mRNAs were mitochondrial proteins (*Figure 3A–B*). The translation of 163 mRNAs of nuclear-encoded mitochondrial proteins was upregulated in Hri$^{-/-}$ −Fe EBs compared to Wt −Fe EBs. These include mitochondrial ribosomal proteins, oxidative phosphorylation proteins (comprising complexes I through V), and matrix proteins (*Figure 3—figure supplement 1C* and *Supplementary file 2b*). Interestingly, this set of mRNAs encoding mitochondrial proteins that are differentially translated in HRI deficiency was distinct from those encoding mitochondrial proteins that are upregulated by mTORC1/eIF4EBPs-mediated translation (*Morita et al., 2013*) (*Figure 3D*). Overall, these genome-wide results indicate roles for both HRI–eIF2αP and mTORC1 signaling in the global translation of ribosomal and mitochondrial proteins in developing EBs during ID.

## Increased cytoplasmic and mitochondrial protein synthesis but reduced mitochondrial respiratory activity in HRI$^{-/-}$ −Fe EBs

To examine the impact of increased translation of ribosomal proteins (both cytosolic and mitochondrial), protein synthesis in the cytoplasm and mitochondria were measured in the primary erythroid cells isolated from bone marrows of both Wt −Fe and Hri$^{-/-}$ −Fe mice. We took advantage of the fact that cycloheximide (CHX) inhibits only cytoplasmic and not mitochondrial protein synthesis. As shown in *Figure 3E*, the rate of total protein synthesis in Hri$^{-/-}$ −Fe erythroid cells at 15–60 mins was greater than that in Wt −Fe cells. Interestingly, the rate of protein synthesis in cells under CHX treatment was also greater in Hri$^{-/-}$ −Fe erythroid cells than in Wt-Fe cells. Furthermore, protein synthesis under CHX treatment was inhibited by chloramphenicol (*Figure 3—figure supplement 2*), a selective inhibitor of mitochondrial translation (*Richter et al., 2013*). These results demonstrate that HRI

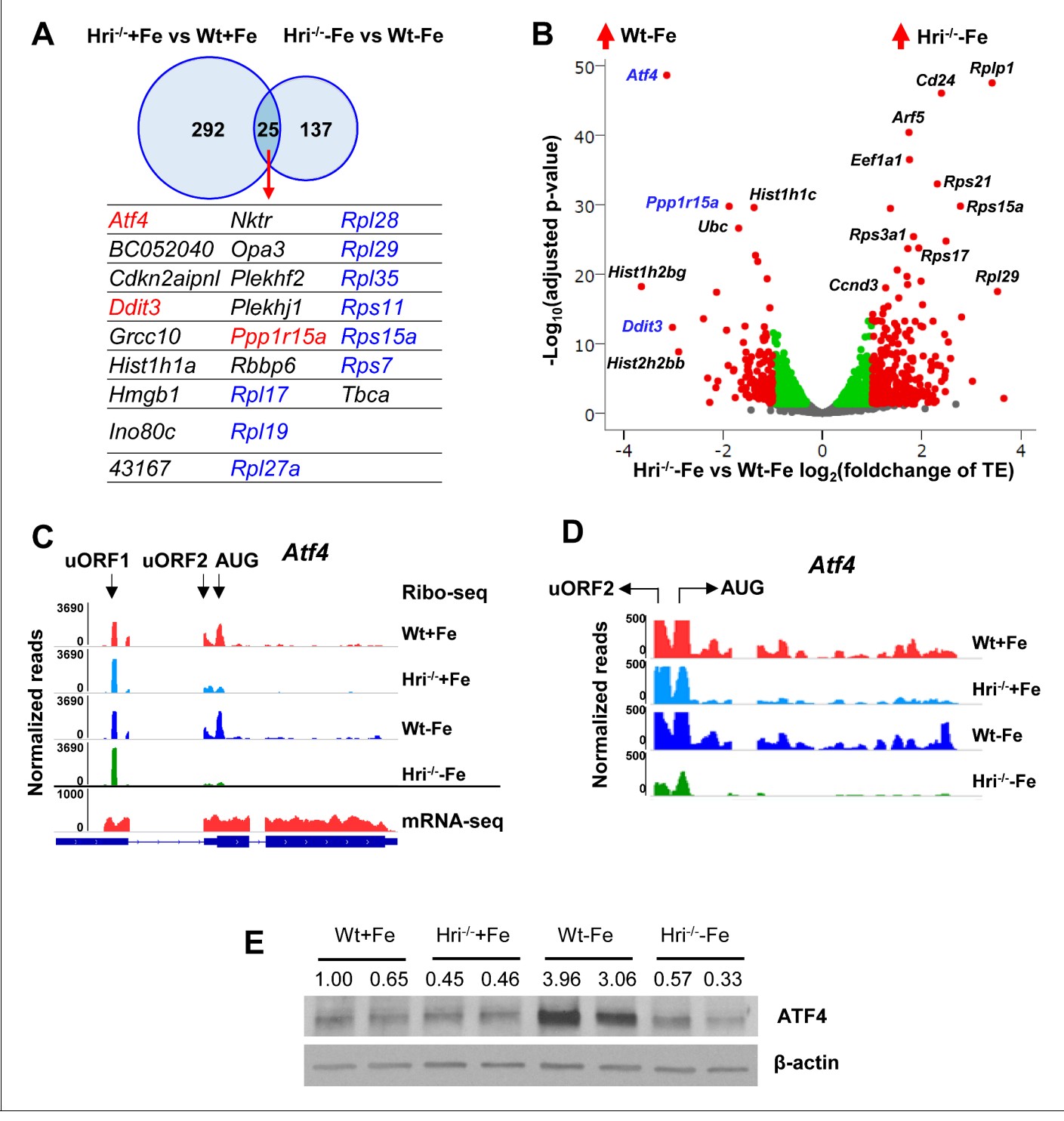

**Figure 2.** Differentially translated mRNAs in Hri[−/−] EBs and iron deficiency. (A) The mRNAs that are significantly differentially translated between Hri[−/−] and Wt EBs in +Fe or –Fe conditions. (B) Volcano plot of mRNAs that are differentially translated between Hri[−/−] –Fe and Wt –Fe EBs. Red dots on the positive end of the X-axis indicate significantly differentially translated mRNAs that were upregulated in Hri[−/−] –Fe EBs, whereas red dots on the negative end of the X-axis indicate significantly differentially translated mRNAs that were upregulated in Wt –Fe EBs. Green and gray dots indicate mRNAs that have no significant difference in translation. TE, translational efficiency. (C) Ribosome occupancies, as visualized using Integrative Genomics Viewer (IGV), of *Atf4* mRNA, with an enlarged view shown in (D). (E) ATF4 protein expression in E14.5 FL cells. Ratios of ATF4/β-actin expression are indicated.

DOI: https://doi.org/10.7554/eLife.46976.004

*Figure 2 continued on next page*

*Figure 2 continued*

The following figure supplement is available for figure 2:

**Figure supplement 1.** Translational regulation of ISR mRNAs by HRI.
DOI: https://doi.org/10.7554/eLife.46976.005

is necessary to inhibit both cytoplasmic and mitochondrial protein synthesis in iron-deficient erythroid cells. To discern the contributions of HRI–eIF2αP and mTORC1 activity in the regulation of mitochondrial protein synthesis, INK128, a mTOR inhibitor, was employed in mitochondrial protein synthesis assays (*Figure 3—figure supplement 3*). INK128 treatment reduced mitochondrial protein synthesis by about 50% with a near complete inhibition of pS6 and p4EBP1. These results show that the HRI–eIF2αP pathway contributes directly to the regulation of mitochondrial protein synthesis during ID, in addition to repressing mTORC1 activity.

We then asked whether the increased protein synthesis in Hri$^{-/-}$ –Fe erythroid cells affects the activity and mass of mitochondria in these cells when compared with Wt –Fe cells. As iron and heme are cofactors for several enzymes in the respiratory electron transport chain, the oxygen consumption rates of primary erythroid cells from bone marrows of +Fe and –Fe mice were measured using the Seahorse assays (*Figure 4*). Wt erythroid cells were able to maintain their oxygen consumption rate in ID (*Figure 4A*). There were no significant differences between Wt –Fe and Wt +Fe cells in either the basal or maximal respiration rates (*Figure 4B*). By contrast, Hri$^{-/-}$ erythroid cells displayed decreased basal and maximal respiration under both +Fe and –Fe conditions, as compared to Wt erythroid cells (*Figure 4A–B*). The similar extent to which respiration was reduced in Hri$^{-/-}$ +Fe cells and Hri$^{-/-}$ –Fe cells was somewhat unexpected because the phenotype of Hri$^{-/-}$+Fe in vivo is milder than that of Hri$^{-/-}$ –Fe cells. This may be due in part to the stress experienced by Hri$^{-/-}$ +Fe cells during the Seahorse assays. Nonetheless, the significant difference between Wt –Fe and Hri$^{-/-}$ –Fe cells demonstrates that HRI is necessary to maintain mitochondrial respiration under –Fe conditions. This impaired mitochondrial respiration in Hri$^{-/-}$ erythroid cells was not due to the reduced mitochondrial mass in these cells (*Figure 4C*). The mitochondrial mass in primary erythroid cells was similar for Wt +Fe and Hri$^{-/-}$ +Fe mice, but was increased in Hri$^{-/-}$ –Fe cells as compared to Wt –Fe cells, especially during the later stages of terminal erythroid differentiation (*Figure 4C*, populations II and III). There was no significant difference in mitochondrial DNA contents between Wt and Hri$^{-/-}$ erythroid cells in either +Fe or –Fe conditions (*Figure 4—figure supplement 1*).

## Cytoplasmic unfolded protein response in Hri$^{-/-}$ –Fe cells

We next investigated the transcriptional impact of ID and the role of HRI in mediating the cellular response to the ID-induced stress response. Analysis of mRNA-seq data revealed that substantially more genes displayed significant differential expression between Wt –Fe and Wt +Fe EBs than between Hri$^{-/-}$ –Fe and Hri$^{-/-}$ +Fe EBs (232 vs 37, *Figure 5A* and *Supplementary file 1e*), demonstrating the near-absolute requirement for HRI in regulating the transcriptional response to ID. GO analysis revealed that the most upregulated process in Hri$^{-/-}$ -Fe EBs when compared to Wt –Fe EBs is the process of protein folding and refolding, which involves many chaperone genes from the *Hsp70* (*Hspa1a, Bag3, Hspa1b, Hspa8, HspA4L*), *Hsp90* (*Hsp90ab1, Hsp90aa1, Chordc1, Ahsa1*), *Hsp40* (*Dnaja4, Dnajb1*), and *Hsp110* (*Hsph1*) families (*Figure 5A–B* and *Figure 6B*). These are cytosolic chaperones that are upregulated specifically in response to cytoplasmic unfolded proteins. Chaperones involved in neither ER (*Hspa5, Hsp90b1*) nor mitochondrial (*Hspa9, Hspd1*) unfolded-protein responses (UPR) were not significantly increased. This comprehensive upregulation of cytoplasmic chaperones indicates the response to cytoplasmic unfolded proteins, such as heme-free globins and nuclear-encoded mitochondrial proteins, in Hri$^{-/-}$ –Fe EBs. Cytoplasmic chaperones were not induced in the presence of HRI during ID (*Figure 5A* and *Figure 6A*), consistent with the essential role of HRI in inhibiting protein synthesis in iron/heme deficiency shown in *Figure 3E*.

We have shown previously that HRI-ISR is activated in an ex vivo FL differentiation system and that erythroid differentiation of Hri$^{-/-}$ progenitors is impaired even under +Fe conditions (*Suragani et al., 2012*). Here, we showed that there was no significant effect of HRI, eIF2αP, or ATF4 deficiencies on growth and viability during expansion and during up to 20 hr of erythroid differentiation (*Figure 5C–D*). However, *eIF2α* Ala51/Ala51 (AA) erythroid precursors, which were

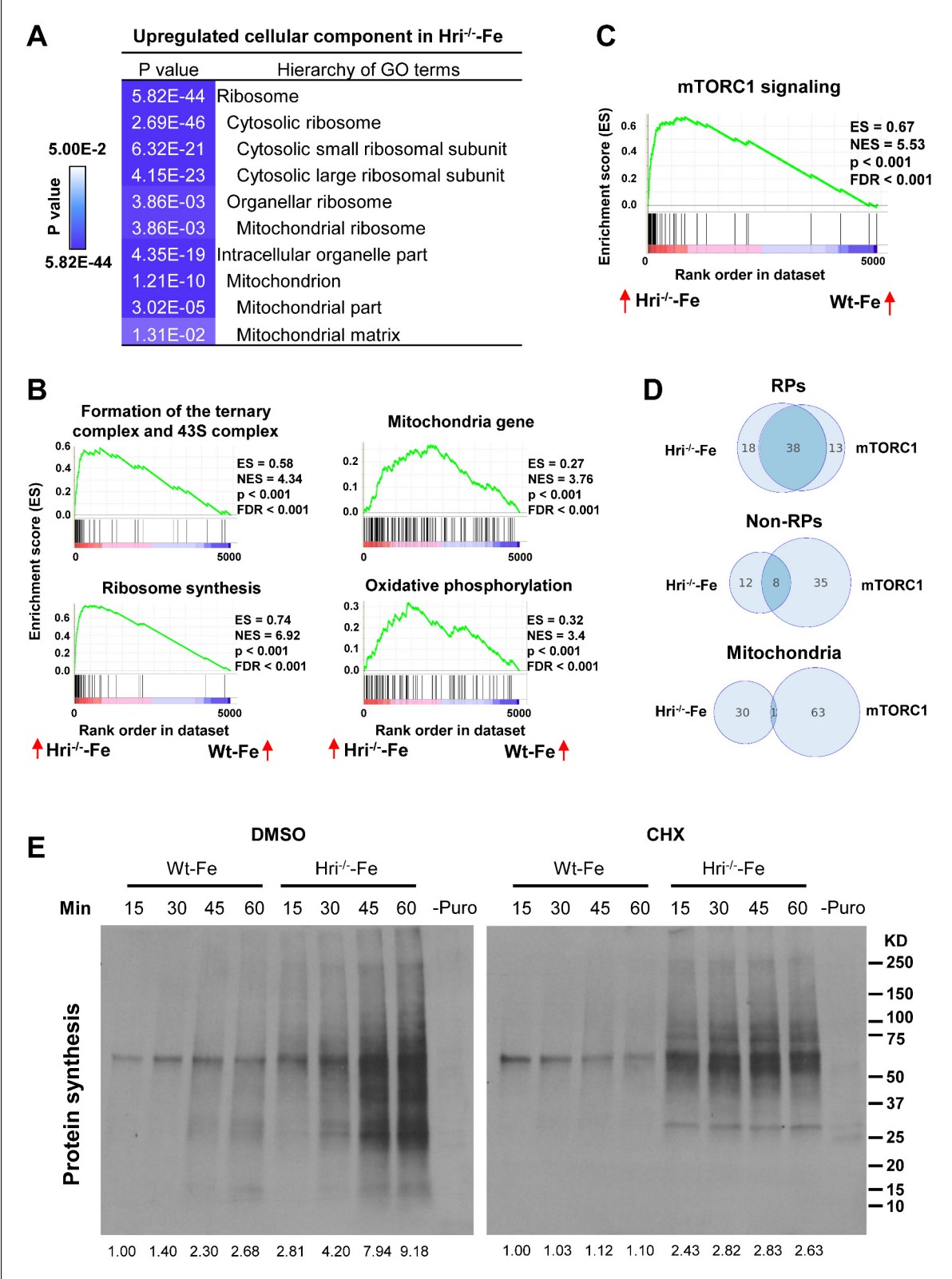

**Figure 3.** Analyses of the mRNAs that are differentially translated between Wt and Hri[−/−] EBs. (**A**) GO analysis of the most highly translated mRNAs in Hri[−/−] –Fe EBs compared to Wt –Fe EBs. (**B**) Increased 43S initiation complex, ribosomal protein (RP) synthesis and mitochondrial pathways in Hri[−/−] –Fe EBs compared to Wt –Fe EBs by Gene Set Enrichment Analysis. (**C**) Increased mTORC1 signaling pathway in Hri[−/−] –Fe EBs compared to Wt –Fe EBs. (**D**) Venn diagrams comparing mRNAs that are differentially translated in Hri[−/−] –Fe EBs and Wt –Fe EBs, and that are known mTORC1 targets in the

*Figure 3 continued on next page*

*Figure 3 continued*

categories of RPs, non-RPs and mitochondrial proteins. (E) Protein synthesis (total (left) and in mitochondria (right)) of erythroid cells from the bone marrow (BM) of Wt –Fe and Hri⁻/⁻ –Fe mice. Cells were treated with dimethyl sulfoxide (DMSO) or cycloheximide (CHX) for 15 min before the addition of puromycin for 15–60 min as indicated. Protein synthesis was determined by the nascent polypeptides covalently linked by puromycin using anti-puromycin antibody. Equal numbers of nucleated cells were loaded to each lane. The numbers of cells loaded for cytoplasmic protein synthesis were 30% of those loaded for mitochondrial protein synthesis. The results shown are from the same exposure time for developing the Western blot. Puromycin signals from polypeptides of the entire lane were quantified for protein synthesis activity and the protein synthesis in the first lane was define as 1. –Puro indicates cells without puromycin treatment, which were used as a negative control for Western signals from anti-puromycin antibody. FDR, false discovery rate; NES, normalized enrichment score.

DOI: https://doi.org/10.7554/eLife.46976.006

The following figure supplements are available for figure 3:

**Figure supplement 1.** Differential translation of cytoplasmic and mitochondrial RP mRNAs between Wt and Hri⁻/⁻ EBs.

DOI: https://doi.org/10.7554/eLife.46976.007

**Figure supplement 2.** Inhibition of mitochondrial protein synthesis by chloramphenicol.

DOI: https://doi.org/10.7554/eLife.46976.008

**Figure supplement 3.** Inhibition of mitochondrial protein synthesis by INK128.

DOI: https://doi.org/10.7554/eLife.46976.009

devoid of eIF2αP, accumulated significant numbers of globin inclusions between 20 hr and 30 hr of erythroid differentiation, resulting in cell death and fragments of cell debris (*Figure 5C*). A similar observation was found in Hri⁻/⁻ erythroid precursors (*Figure 5C*), but the effect was less severe in these cells because eIF2αP is completely absent from AA erythroid precursors, whereas Hri⁻/⁻ cells still have a low level of eIF2αP.

Together, the presence of aggregated protein inclusions in ex vivo differentiation and the induction of cytoplasmic UPR in Hri⁻/⁻ EBs underscore the primary function of HRI–eIF2αP in inhibiting translation, and thereby coordinating protein homeostasis with available iron and heme concentrations to mitigate proteotoxicity during FL erythropoiesis.

## HRI-ATF4 mediated mRNA expression is most highly upregulated in ID

As shown in *Figure 5D*, Atf4⁻/⁻ FL erythroid cells did not suffer from proteotoxicity due to the presence of functional HRI-eIF2αP in inhibiting globin mRNA translation (*Zhang et al., 2018*). However, the differentiation of Atf4⁻/⁻erythroid precursors at 30 and 42 hr was impaired (*Figure 5D*). These results demonstrate that the enhanced translation of Atf4 mRNA is also required for erythroid differentiation.

The majority of highly induced genes in Wt –Fe EBs compared to Wt +Fe EBs were ATF4 target genes (*Figure 6A*) (*Pakos-Zebrucka et al., 2016*). However, these genes were not upregulated in Hri⁻/⁻ –Fe EBs (*Figure 6B*, *Figure 6—figure supplement 1A*), indicating that HRI is required to activate ISR in ID (*Figure 2*). The GO terms for the biological processes that were most highly differentially expressed in Wt –Fe EBs as compared to Hri⁻/⁻ –Fe EB were amino acid metabolism and biosynthesis (*Figure 6C*). The corresponding upregulated ISR genes are involved in serine-glycine biosynthesis (*Phgdh*, *Psat1*, *Psph*, *Shmt2*), one-carbon metabolism (*Shmt2*, *Mthfd2*), proline-glutamine synthesis (*Pycr1*, *Aldh18a1*, *Asns*) and glutathione metabolism (*Chac1*) (*Figure 6B*, *Figure 6—figure supplement 1B* and *Supplementary file 1e*). Furthermore, the expression levels of these genes were lower in Hri⁻/⁻ EBs than in Wt EBs under the +Fe condition (*Figure 6—figure supplement 1C*), suggesting that HRI finetunes the ISR during iron replete erythropoiesis and thus has an important role even under normal conditions. Interestingly, some of these ATF4 target genes, most notably *Atf5*, *Trib3* and *Chac1*, are also Epo-stimulated genes (*Figure 6—figure supplement 1D*) (*Singh et al., 2012*), consistent with the interaction of the HRI-ISR pathway and Epo signaling (*Zhang et al., 2018*).

## *Grb10* is required for late-stage erythroid differentiation

Having mapped the genome-wide translational and transcriptional responses to ID, we set out to characterize the function of one of the most highly induced ATF4 target genes, growth factor receptor-bound protein 10 (*Grb10*), in erythropoiesis. *Grb10* has been reported to be part of a feedback mechanism that inhibits the mediation of mTORC1 signaling by growth factors such as insulin

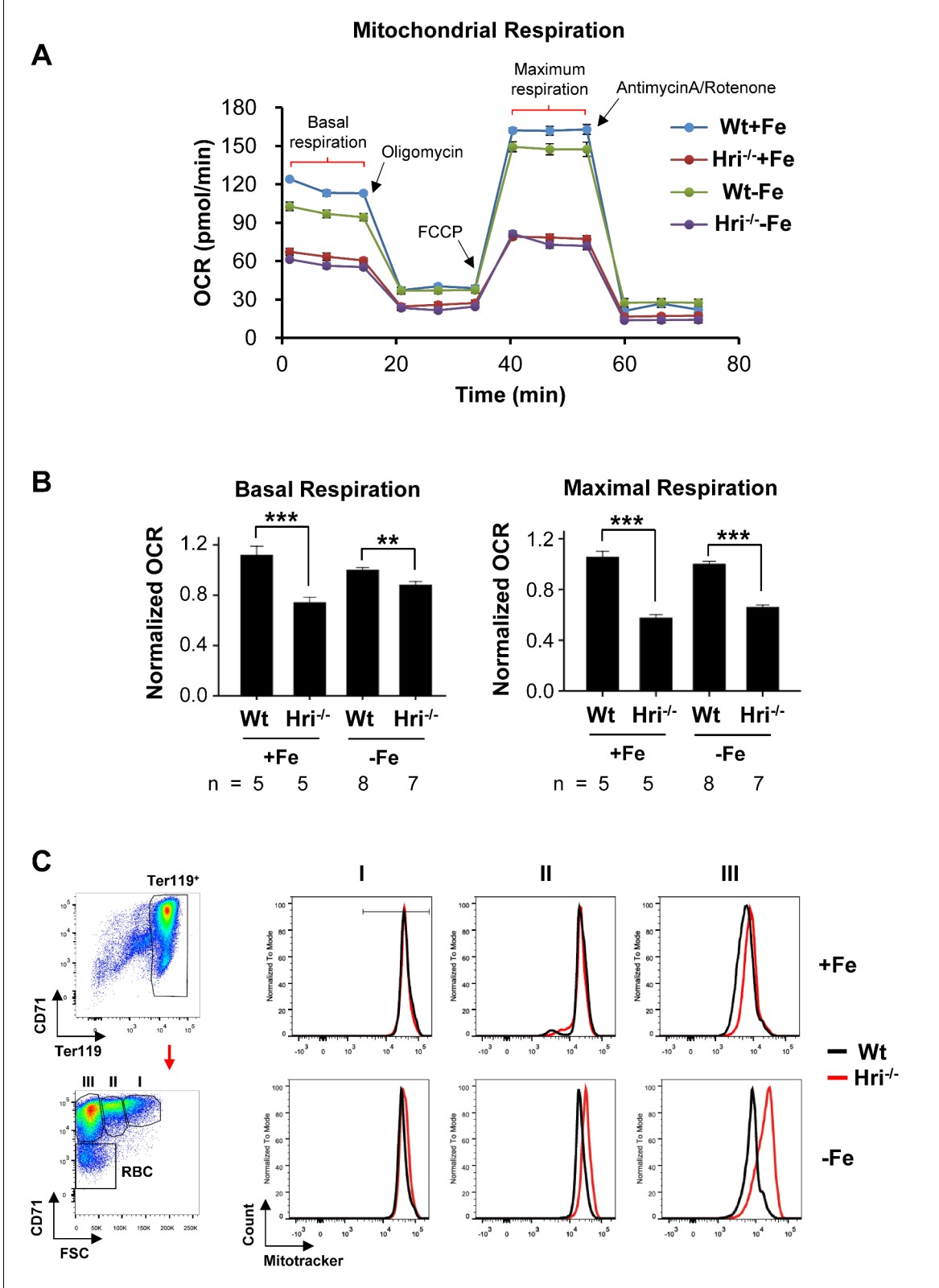

**Figure 4.** Decreased mitochondrial respiration in HRI deficiency. (**A**) A representative result for the oxygen consumption rate (OCR) of Wt and Hri$^{-/-}$ erythroid cells. Erythroid cells were isolated from the BM of Wt and Hri$^{-/-}$ mice in +Fe and −Fe conditions. Five technical replicas were performed for each condition. (**B**) Quantitative analysis of basal and maximal OCR from four separate experiments. The numbers of mice used in each condition are indicated. OCR of Wt −Fe erythroid cells was defined as 1. Data were presented as mean ± SE. **p<0.01, ***p<0.001. (**C**) Representative plots of
*Figure 4 continued on next page*

*Figure 4 continued*

mitochondrial mass of erythroid cells at different differentiation stages obtained from mitotracker flow cytometry analysis. A diagram of gating for the differentiation stages is shown on the left.

DOI: https://doi.org/10.7554/eLife.46976.010

The following figure supplement is available for figure 4:

**Figure supplement 1.** The relative mitochondrial DNA contents of erythroid cells.

DOI: https://doi.org/10.7554/eLife.46976.011

(*Plasschaert and Bartolomei, 2015*) and stem cell factor (SCF) (*Yan et al., 2016*). We employed ex vivo FL differentiation to interrogate the function of *Grb10* in erythropoiesis.

First, we validated that *Grb10* and *Atf5*, but not *Atf4*, expression was increased in Wt –Fe EBs, but not in Hri$^{-/-}$ –Fe EBs (*Figure 7—figure supplement 1A*). In addition, *Grb10* expression in Wt EBs was increased during ex vivo erythroid differentiation from 20 to 30 hr, and was greatly reduced in Hri$^{-/-}$, AA and *Atf4$^{-/-}$* erythroblasts (*Figure 7—figure supplement 1B*). We prepared eight shRNA recombinant retroviruses, all of which were able to knockdown *Grb10* RNA expression by more than 80% during the expansion phase. Two of these retroviruses, shRNA_G3 and shRNA_G7, were able to maintain persistent knockdown of *Grb10* RNA during differentiation (*Figure 7A*, *Figure 7—figure supplements 2–3*). Reduction of *Grb10* expression by shRNAs (*Figure 7D* and *Figure 7—figure supplement 3A*) increased the numbers of differentiating erythroblasts (*Figure 7B*). This increase was mostly observed between 16 and 26 hr of differentiation (*Figure 7B*), a period during which Epo concentrations were increased and SCF was withdrawn (*Figure 7A*). We also observed increased expression of cyclin D3, a known target of mTORC1 signaling (*Figure 7D*). Importantly, terminal erythroid differentiation was inhibited in *Grb10* knockdown cells, as indicated by an accumulation of polychromatic erythroblasts and a decrease in orthochromatic erythroblasts (*Figure 7C*). Thus, the increased proliferation but decreased terminal differentiation upon reduction of *Grb10* expression (*Figure 7*) recapitulates the hallmarks of ineffective erythropoiesis observed in Hri$^{-/-}$ mice in ID (*Han et al., 2001*; *Suragani et al., 2012*; *Zhang et al., 2018*).

These results strengthen our global observation of an interaction between HRI-ISR and the Epo signaling pathway by showing that the induction of *Grb10* by HRI-ISR serves as an important feedback mechanism for Epo signaling, resulting in reduced proliferation and thus promoting erythroid differentiation during stress erythropoiesis (*Figure 8*).

## Discussion

Although the specific role of HRI in the translational regulation of globin and *Atf4* mRNAs in erythroid cells has been appreciated, an understanding of the global impact of HRI-mediated translational regulation on erythropoiesis is lacking. Here, we report a global, unbiased assessment of iron, heme, and HRI-mediated translational and transcriptional alterations in primary EBs in vivo. We show that *Eif2ak1* and *Atf4* mRNAs are abundantly expressed in EBs and are poised to respond to ID during terminal erythroid differentiation. HRI is a major regulator of gene expression in erythropoiesis: only limited changes of mRNA expression are observed in Hri$^{-/-}$ EBs during ID. Our global analysis supports a distinct model of heme and HRI-regulated translational control involving several separate and interacting pathways that allow EBs to adapt to systemic ID (*Figure 8*). Activation of HRI in ID elicits three distinct pathways of translation. First and foremost, inhibition of protein synthesis prevents the accumulation of unfolded proteins and maintains protein homeostasis. Second, induction of *Atf4* mRNA translation increases the expression of genes encoding mitochondrial UPR, redox homeostasis and metabolic reprogramming in order to maintain mitochondrial respiration and erythroid differentiation. Third, repression of Epo-mTORC1 signaling inhibits protein synthesis and enables erythroid differentiation.

The present study identifies a previously unappreciated role of HRI in the translational regulation of cytosolic and mitochondrial ribosomal proteins during erythropoiesis. We observed, at a global level, additional evidence of mTORC1's involvement in translational regulation in Hri$^{-/-}$ –Fe EBs, further supporting the role of HRI-ISR in repressing Epo–mTORC1 signaling to mitigate ineffective erythropoiesis during ID (*Zhang et al., 2018*). Importantly, we find that HRI–eIF2αP also controls the translation of 18 cytosolic RP mRNAs, which are not known targets of mTORC1. Consistent with our

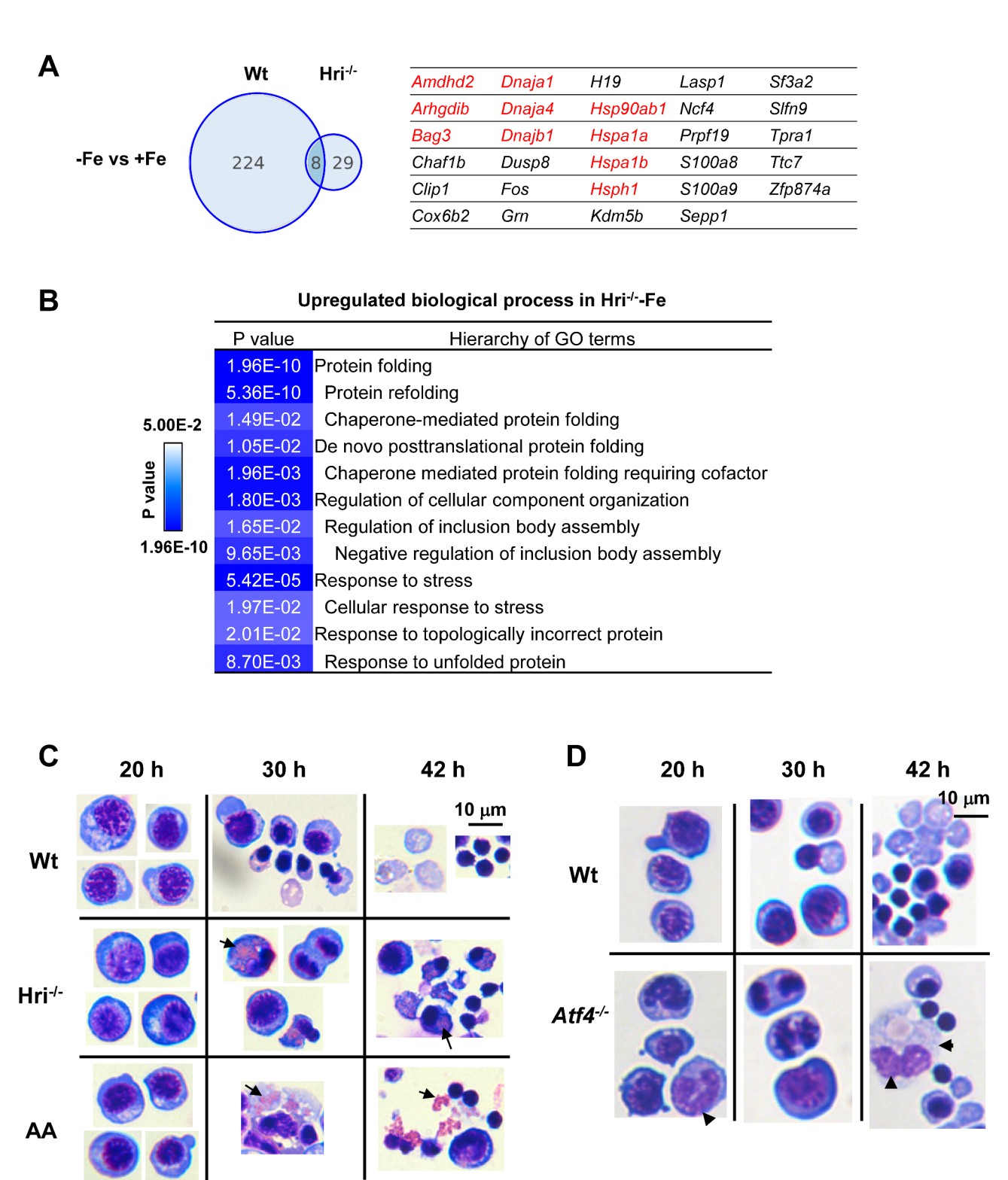

**Figure 5.** Increased expression of cytoplasmic chaperones, accumulation of unfolded protein inclusions and impaired FL differentiation in Hri⁻/⁻ erythroid cells. (**A**) Venn diagrams depicting the numbers of mRNAs that are significantly differentially expressed between –Fe and +Fe conditions in Wt or Hri⁻/⁻ EBs. The table on the right lists mRNAs that were expressed at higher level only in Hri⁻/⁻ –Fe EBs. Results were obtained from three biological replicas. (**B**) GO terms related to protein folding which were most highly upregulated in Hri⁻/⁻ –Fe EBs compared to Wt –Fe EBs from the Gene

*Figure 5 continued on next page*

*Figure 5 continued*
ontology analysis of significantly differentially expressed mRNAs. (**C–D**) Ex vivo erythroid differentiation from HRI-, eIF2αP-, and ATF4-deficient FL erythroid progenitors. The representative images are of cytospin slides stained with May-Grunwald/Giemsa staining. Cells at 20, 30 and 42 hr of erythroid differentiation are shown. AA, universal *eIF2α* Ala51/Ala51 knockin resulting in complete ablation of eIF2αP. Arrow in (**C**) indicates globin inclusions and arrowhead in (**D**) indicates myeloid cells. Numbers of FL differentiation performed were: n = 6 for Wt and Hri⁻ᐟ⁻; n = 4 for AA and n = 3 for *Atf4*⁻ᐟ⁻. Some of the cell images in (**C**) were cropped from the same slides and compiled together for illustrative purposes.
DOI: https://doi.org/10.7554/eLife.46976.012

results, mTORC1-independent translation of RP mRNAs requiring GCN2-eIF2αP has been reported recently in several cancer cell lines (*Li et al., 2018*). Interestingly, the translation of many mitochondrial RPs and oxidative phosphorylation complexes is also increased in Hri⁻ᐟ⁻ –Fe EBs. Furthermore,

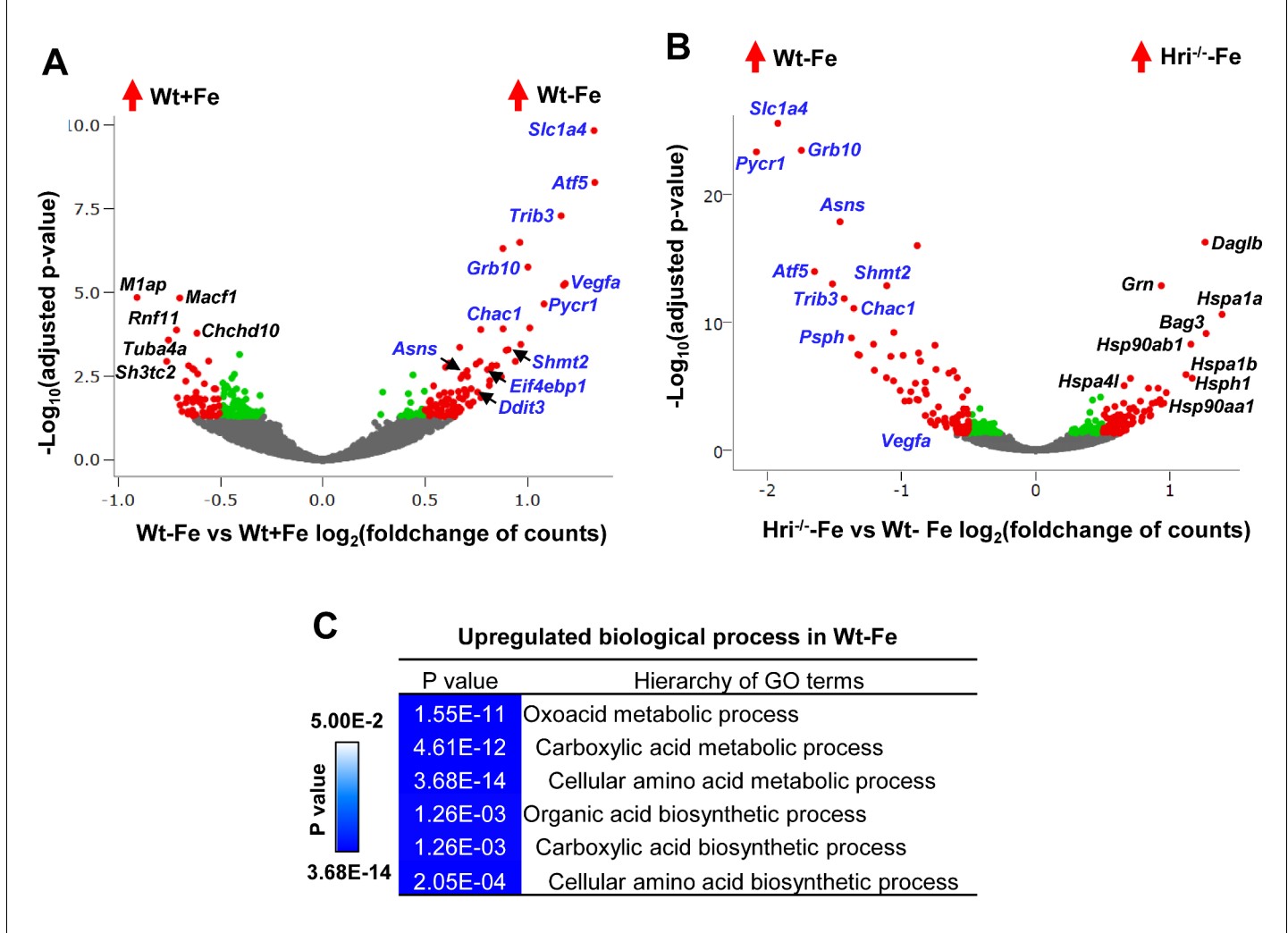

**Figure 6.** Analyses of differentially expressed mRNAs in iron and HRI deficiencies. (**A**) Volcano plots of mRNAs that are differentially expressed between Wt –Fe and Wt +Fe or (**B**) between Hri⁻ᐟ⁻ –Fe and Wt –Fe EBs. Red dots represent the significantly differentially expressed mRNAs. Green and gray dots indicate mRNAs whose expression does not differ significantly. ATF4 target genes are labeled in blue. (**C**) GO terms for amino acid metabolism that are most highly upregulated in Wt –Fe EBs compared to Hri⁻ᐟ⁻ –Fe EBs.
DOI: https://doi.org/10.7554/eLife.46976.013
The following figure supplement is available for figure 6:

**Figure supplement 1.** Analyses of differentially expressed ISR-target mRNAs.
DOI: https://doi.org/10.7554/eLife.46976.014

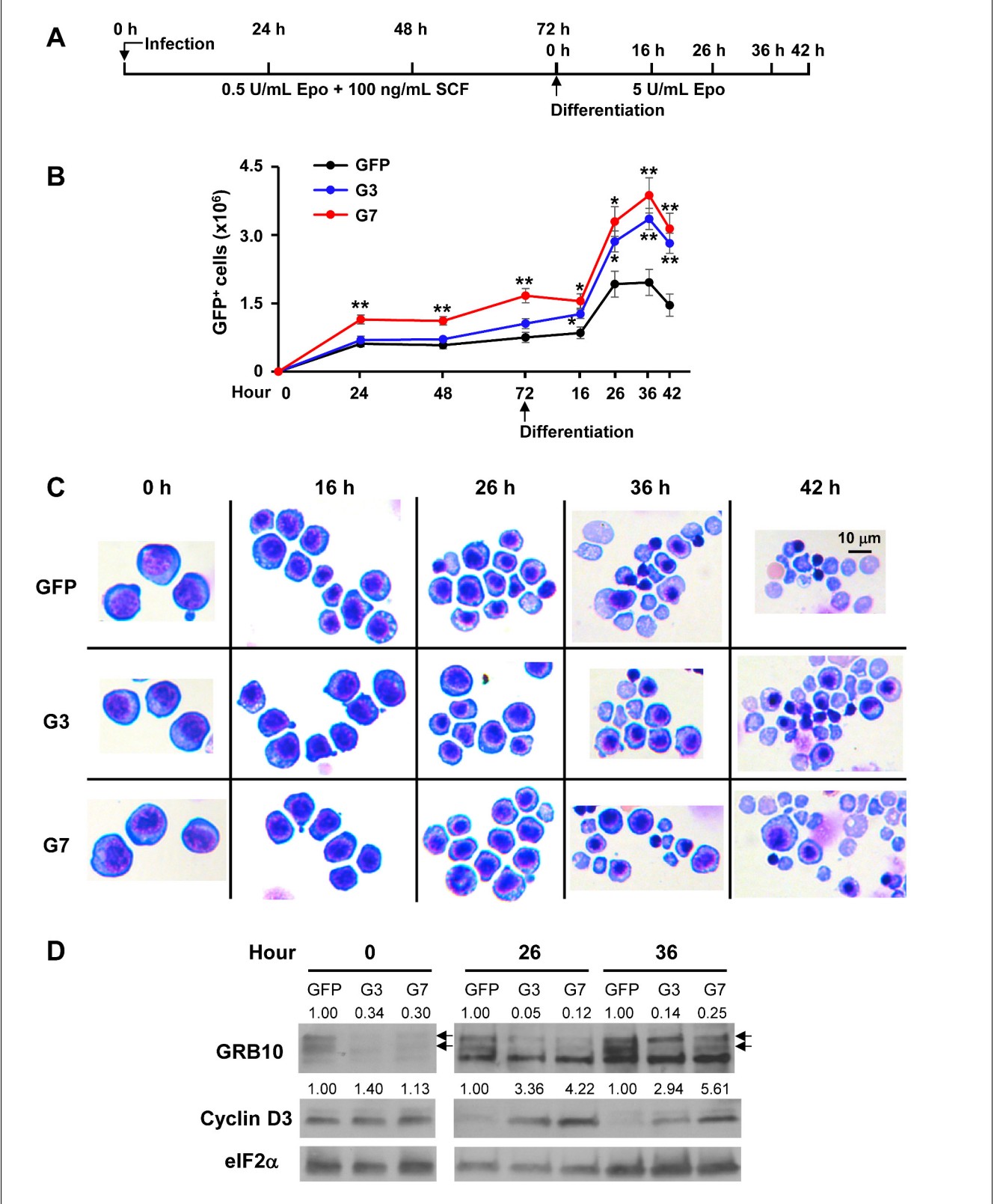

**Figure 7.** Knockdown of *Grb10* increases proliferation and inhibits differentiation of FL erythroblasts. (A) Designs of shRNA knockdown experiments using Lin⁻Ter119⁻CD71⁻ FL erythroid progenitors. (B) Proliferation of control and *Grb10* knockdown GFP⁺ cells. Three biological replicas were performed. Data are presented as means ± SEs. *p<0.05, **p<0.01. (C) Representative cell morphology of GFP control and *Grb10* knockdown cells. (D) GRB10 and Cyclin D3 protein levels of GFP control and *Grb10* knockdown cells during differentiation. Arrows indicate the doublets of GRB10. The
*Figure 7 continued on next page*

*Figure 7 continued*

green fluorescent protein (GFP) control was infected with retrovirus expressing GFP only; G3 and G7 were infected with retroviruses expressing shRNA_G3 and shRNA_G7, respectively.

DOI: https://doi.org/10.7554/eLife.46976.015

The following figure supplements are available for figure 7:

**Figure supplement 1.** Expression of *Atf4*, *Atf5* and *Grb10* mRNAs in HRI-ISR defective erythroid cells.

DOI: https://doi.org/10.7554/eLife.46976.016

**Figure supplement 2.** Enrichment of Lin⁻Ter119⁻CD71⁻ erythroid progenitors from Wt +Fe E13.5 FLs for ex vivo experiments.

DOI: https://doi.org/10.7554/eLife.46976.017

**Figure supplement 3.** Knockdown efficiency of *Grb10* in FL cells during ex vivo differentiation.

DOI: https://doi.org/10.7554/eLife.46976.018

HRI directly inhibits mitochondrial protein synthesis in addition to repressing mTORC1 activity. Thus, both the HRI–eIF2αP and the mTORC1 pathways contribute to the increased mitochondrial protein synthesis in Hri⁻/⁻ –Fe EBs.

Increased protein synthesis in Hri⁻/⁻ –Fe EBs requires additional protein folding capacity for heme-free globins and increased import into mitochondria of mitochondrial protein precursors synthesized in the cytoplasm. Indeed, Hri⁻/⁻ –Fe EBs develop a prominent cytoplasmic UPR, as suggested here by the increased expression of cytoplasmic chaperones in these cells. These results are consistent with and help to identify the mechanisms underlying our previous reports of insoluble inclusions and protein precipitates accompanied by increased levels of Hsp70 and Hsp90 in erythroid cells of Hri⁻/⁻ and *eAA* mice in ID (*Han et al., 2001*; *Zhang et al., 2018*). Nevertheless, activation of cytoplasmic UPR alone is not sufficient to compensate for the loss of HRI.

Translational regulation of mitochondrial biogenesis by mTORC1 is particularly important for erythropoiesis because of the high demand for heme for hemoglobin production and the prevention of oxidative stress (*Liu et al., 2017*). Most mitochondrial proteins are encoded by nuclear genes and synthesized by cytosolic ribosomes as precursors that are imported into mitochondria. Mitochondrial DNA encodes only 13 proteins, which are all components of the respiratory chain and oxidative phosphorylation (*Calvo and Mootha, 2010*). Therefore, the nuclear and mitochondrial synthesis of proteins that are involved in respiratory complexes needs to be coordinated to avoid excessive uncomplexed subunits (*Priesnitz and Becker, 2018*). In mitochondrial disorders, mitochondrial UPR (UPRᵐᵗ) is activated to coordinate with nuclear transcription in order to enable restoration of mitochondrial function (*Shpilka and Haynes, 2018*). Interestingly, ISR is also activated in mitochondrial dysfunction and mediates UPRᵐᵗ (*Fiorese et al., 2016*; *Quirós et al., 2017*; *Dogan et al., 2014*). However, the eIF2α kinase responsible for UPRᵐᵗ remains unknown. Our observations that Hri⁻/⁻ erythroid cells have impaired mitochondrial function and no significant increase in the expression of mitochondrial chaperones suggest that HRI may be the eIF2α kinase responsible for the activation of UPRᵐᵗ and might thereby contribute to the proper maintenance of the mitochondrial function of erythroid cells under ID.

While inhibition of protein synthesis by the activation of HRI is the first line of defense in reducing proteotoxicity in ID, the second arm of the HRI-ISR, inducing ATF4 target gene expression as a consequence of enhanced translation of *Atf4* mRNA, is the most highly activated transcriptome pathway in ID. These ATF4 target genes are largely unstudied factors with yet unknown functions in erythropoiesis. They are involved in redox and amino acid metabolism, particularly one-carbon metabolism. Interestingly, these ATF4-ISR genes are also upregulated under mitochondrial stress (*Quirós et al., 2017*; *Bao et al., 2016*) in order to maintain mitochondrial redox homeostasis. The deficiency of induction of ATF4-ISR gene expression is likely to be responsible for elevated oxidative stress in erythroid cells from iron-deficient Hri⁻/⁻, eAA and *Atf4⁻/⁻* mice (*Suragani et al., 2012*; *Zhang et al., 2018*), as well as the mitochondrial dysfunction reported here.

Hyporesponsiveness to Epo therapy is commonly encountered in ID (*Goodnough, 2007*). Our recent studies provide evidence that HRI-ISR constitutes one feedback mechanism of erythroid homeostasis in ID (*Zhang et al., 2018*). We show here that *Grb10* is probably one of the ATF4 target genes involved in inhibiting Epo–mTORC1 signaling in ID. The increased proliferation of differentiating erythroblasts upon knockdown of *Grb10* is consistent with the growth suppressor function of *Grb10* (*Plasschaert and Bartolomei, 2015*). In hematopoiesis, *Grb10* regulates the self-renewal and

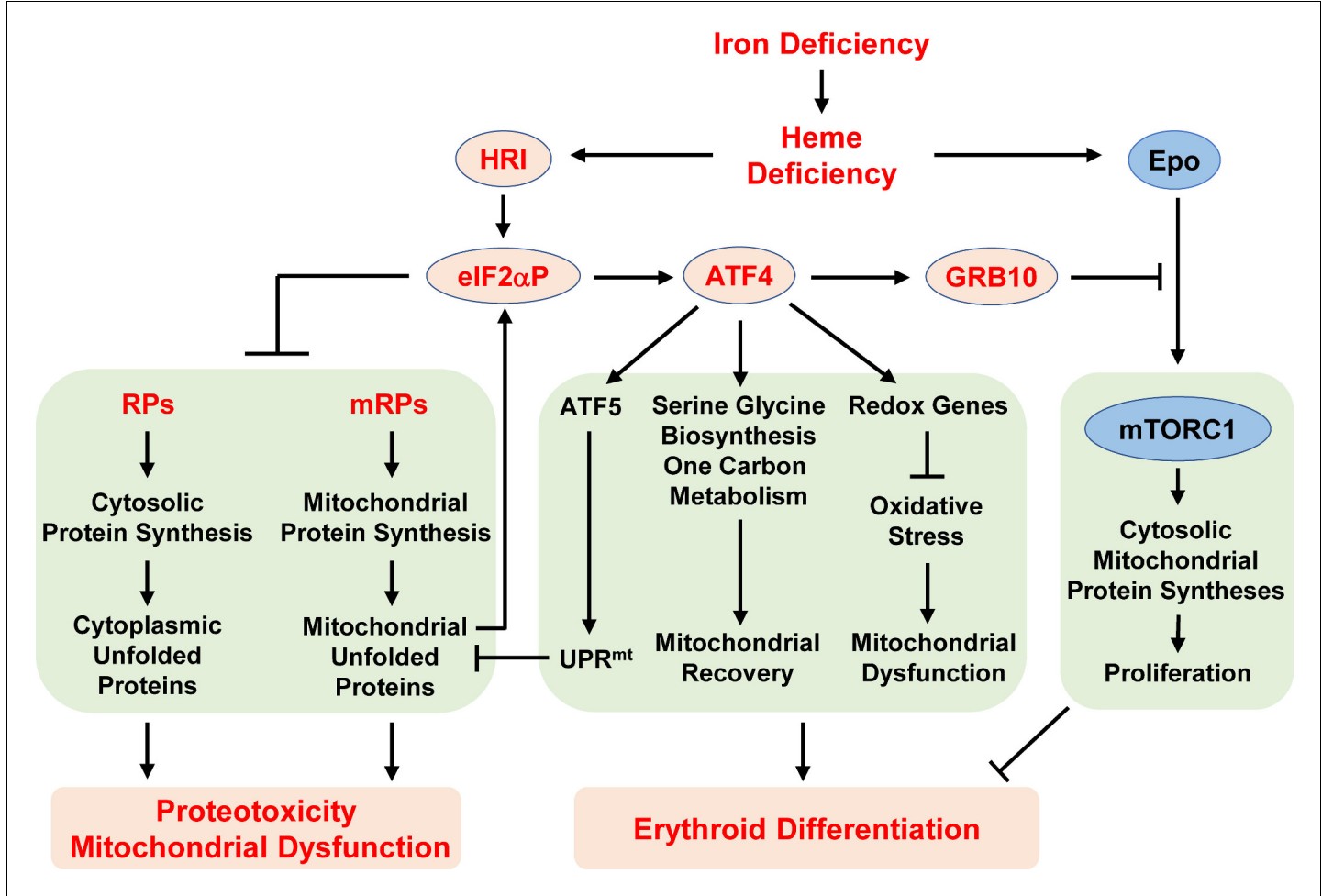

**Figure 8.** Proposed model of the regulation of erythropoiesis by heme and the HRI-ISR pathway during ID. Diet-induced systemic ID results in heme deficiency, which activates HRI to phosphorylate its substrate eIF2α. The primary function of eIF2αP is to inhibit general protein synthesis in both the cytoplasm and mitochondria via decreased translation of ribosomal protein mRNAs from both cellular compartments. In the absence of HRI, continued protein synthesis results in the accumulation of unfolded proteins in both the cytoplasm and mitochondria, leading to proteotoxicity and mitochonfrial dysfunction. Second, eIF2αP also enhances the translation of *Atf4* mRNA via regulation by uORFs. *Atf4* mRNA is the mRNA that shows the greatest differences in translation levels between Wt and Hri$^{-/-}$ EBs. The ATF4 protein induces the expression of three genetic pathways to activate mitochondrial UPR (UPR$^{mt}$), reprogram mitochondrial metabolism and reduce oxidative stress, all of which enables adaptation to ID and erythroid differentiation. Strikingly, the expression of ATF4 target genes is most highly activated in ID and requires HRI. Thus, assessment of the global genome-wide gene expression of primary EBs in vivo reveals that HRI-ISR contributes most significantly to adaptation to ID. We have shown previously that HRI-ISR suppressed mTORC1 signaling, which is activated by elevated Epo levels in ID (*Zhang et al., 2018*). Here, we provide evidence that GRB10 that is induced by ATF4 may be one of the molecules involved in the repression of mTORC1 signaling.

DOI: https://doi.org/10.7554/eLife.46976.019

The following figure supplement is available for figure 8:

**Figure supplement 1.** Expression of *Atf4* mRNA in hematopoietic cells.

DOI: https://doi.org/10.7554/eLife.46976.020

regeneration of hematopoietic stem cells (HSCs). *Grb10*-deficient HSCs exhibit increased proliferation that is dependent on the SCF-AKT/mTORC1 pathway (*Yan et al., 2016*), consistent with our *Grb10* knockdown results. *Grb10* has also been shown to be a GATA2 target gene and to play a role in the transition from BFU-E to CFU-E (*Mehta et al., 2017*). Our results support an additional role of ATF4-induced *Grb10* expression in terminal erythropoiesis in which GATA2 expression is low.

 Notably, *Atf4* mRNA is most highly expressed in Ter119$^+$CD71$^+$ erythroblasts among the 16 differentiation stages of murine bone marrow hematopoietic cells (*Figure 8—figure supplement 1*) (*Lara-Astiaso et al., 2014*). Altogether, our findings underscore the role of HRI-mediated translation

of *Atf4* mRNA in terminal erythroid differentiation. In summary, our genome-wide study reveals the prominent and coordinated contribution of heme- and HRI-mediated translational regulation in maintaining protein homeostasis, mitochondrial activity and erythroid differentiation during iron-restricted erythropoiesis.

# Materials and methods

## Animals and diet-induced iron deficiency

Mice were maintained at the Massachusetts Institute of Technology (MIT) animal facility, and all experiments were carried out using protocols approved by the Division of Comparative Medicine, MIT. $Hri^{-/-}$, $Atf4^{-/-}$ and universal $eIF2\alpha$ $Ala51/+$ heterozygote knockin mice were as described previously (*Han et al., 2001*; *Masuoka and Townes, 2002*; *Scheuner et al., 2001*). Diet-induced iron deficiency in mice was achieved as previously described (*Han et al., 2001*; *Zhang et al., 2018*). Iron-sufficient and -deficient mice (8–12 weeks old) were mated for E13.5 or E14.5 fetal livers as sources of erythroid precursors. Under these iron-deficient conditions, embryos were pale and anemic with decreased hematocrits in embryonic blood (*Liu et al., 2008b*).

## Isolation of EBs, library preparations and DNA sequencing

EBs from E14.5 FLs of Wt and $Hri^{-/-}$ mice maintained under +Fe or –Fe conditions were sorted using anti-Ter119 and anti-CD71 antibodies by flow cytometry using FACS Aria (BD Biosciences, San Jose, CA) (*Figure 1A*). In order to have sufficient EBs for Ribo-seq library, FLs from embryos of the same mother were pooled then sorted for EBs as one biological replica. Two (5 million cells each) and three biological replicas (1 million cells each) for each condition from separate mothers were collected for preparation of Ribo-seq and mRNA-seq libraries, respectively. The third replica of Ribo-seq using 3 million cells was unsuccessful, probably because of lower cell numbers. All procedures for labeling and washing of cells for sorting were carried out at 4°C. Cells were sorted into tubes with 20% fetal bovine serum (FBS, Atlanta Biologicals, Norcross, GA) to help to preserve cell integrity.

To preserve polyribosomes, sorted EBs were washed twice with cold phosphate-buffered saline (PBS) and then treated with cyclohex123imde (100 µg/mL) at 37°C for 5 min, followed by isolation of RPFs as previously described (*Guo et al., 2010*; *Ingolia et al., 2011*) using the ARTseq-Ribosome Profiling Kit (Illumina, San Diego, CA). Briefly, cells were lysed in cold polysome buffer on ice for 10 min. To generate RPFs, lysates were digested with Rnase 1. Monosomes were then purified by S-400 gel filtration spin columns (GE Healthcare, Chicago, IL). RNAs in monosomes were extracted and purified. Size selection of 28–34 nucleotide RPFs was carried out using denaturing polyacrylamide gel electrophoresis. Ribosomal RNA contaminants in the RPF preparations were removed using an Ribo-Zero rRNA Removal Kit (Illumina, San Diego, CA). RPFs were then purified by 15% urea-polyacrylamide gel electrophoresis. Libraries of RPFs were prepared as described previously (*Ingolia et al., 2009*; *Ingolia et al., 2011*) for Illumina next generation DNA sequencing. RPFs were dephosphorylated and linkers were ligated using T4 RNA ligase. RNA samples were then reverse transcribed, circularized and PCR amplified for 12 cycles. PCR products were subjected to gel purification before sequencing. The results of Ribo-seq presented in *Figure 2* were from the second set of experiments. For the first set of Ribo-seq experiments, EBs were treated with harringtonine (2 µg/ml) at 37°C for 2 min followed by cycloheximide treatment as described above. However, harringtonine treatment at this condition did not work well for EBs and consequently ribosome occupancy was observed throughout many mRNAs. For the second set of Ribo-seq experiments, harringtonine was not used. Nonetheless, increased translation of *Atf4*, *Pppr115a* and *Ddit3* mRNAs was also observed in both sets of Ribo-seq.

Total RNAs were extracted using an RNeasy Plus kit (Qiagen, Germantown, MD) and polyA$^+$ mRNAs were isolated using an Oligotex mRNA kit (Qiagen, Germantown, MD). mRNA-seq cDNA libraries were prepared by the MIT BioMicro Center. cDNA libraries of RPFs and mRNA were sequenced on an HiSeq 2000 platform (Illumina, San Diego, CA) at the MIT BioMicro Center. After standard preprocessing and quality control analysis, reads were mapped to mouse genome mm10 (UCSC), beforedownstream analyses were performed.

## Genome-wide data analysis

Raw data (fastq files) were trimmed by Cutadapt to remove adapters and reads with a base quality score of less than 10. Reads with a length of less than 26 or 9 nucleotides were also discarded for Ribo-seq and mRNA-seq samples, respectively. For Ribo-seq samples, reads containing rRNA and tRNA sequences were further removed using Bowtie2. After quality-control analysis using FastQC, reads were mapped to mouse genome mm10 (UCSC) using STAR aligner with maxima of two mismatches and eight multiple loci. The quality of the Ribo-seq data was examined by the triplet periodicity using RibORF (*Ji et al., 2015*).

For the visualization of ribosome occupancies, all mapped reads (bam files) that overlap each bin (bin size = 1) were first calculated and normalized to effective mouse genome size to get a 1x depth of coverage (RPGC) using bamCoverage of deepTools (*Reimand et al., 2016*; *Ji et al., 2015*). Then, ribosome occupancies were visualized using the Integrative Genomics Viewer (Broad Institute of MIT and Harvard). As there was no significant change in mRNA levels among four conditions, only mapped reads of Wt + Fe EBs are shown for mRNA-seq data in *Figure 2C–D*, *Figure 2—figure supplement 1B–C* and *Figure 3—figure supplement 1B–C*.

The uniquely mapped reads were counted using HTseq for gene coverage analysis, transcript per million (TPM) calculation, translational efficiency (TE) calculation and analysis of differentially expressed mRNAs (DEG). Genes with greater than 25 uniquely mapped reads in at least one of the conditions were used for gene coverage analysis and TPM calculation.

Generally, RPFs are piled up around the start codons and stop codons because of the slower kinetics of translation initiation and termination. The use of cycloheximide to freeze polysomes during the preparation of RPFs also enhances the pile up of reads near start codons. We therefore removed the first 15 and the last 5 codons from reads counting for the TE calculation of Ribo-seq data, in which TE is defined as the ratio of RPF counts to mRNA counts. TE was calculated using the xtail package of R/Bioconductor, with parameter 'bins = 10000', in order to identify genes that are differentially translated between different conditions (*Xiao et al., 2016*). Genes with reads greater than 50 in at least one of the conditions were included in each comparison and were used for TE calculation (*Supplementary file 1b*). Genes with adjusted p-value<0.05 together with $\log_2$(foldchange of TE)>1.5 or <−1.5 were considered to be significantly differentially translated (*Figure 2A*), whereas those with adjusted p-value<0.05 together with $\log_2$(foldchange of TE)>1 or < −1 were considered to be significantly differentially translated (*Figure 2B*, *Figure 3A*, *Figure 2—figure supplement 1A* and *Figure 3—figure supplement 1A*).

For Gene Set Enrichment Analysis (GSEA), pre-ranked gene lists were obtained using the formula –$\log_{10}$(adjusted p-value) $\times$ $\log_2$(foldchange of TE). Mouse gene symbols were converted into human symbols using the biomaRt package of R/Bioconductor. GSEA was conducted using the GSEA tool from the Broad Institute with the c2.all.v6.0.symbols.gmt database, which includes 4738 gene sets. Gene sets shown in *Figure 3B–C* were REACTOME_FORMATION_OF_THE_TERNARY_COMPLEX_AND_SUBSEQUENTLY_THE_43S_COMPL, KEGG_RIBOSOME, WONG_MITOCHONDRIA_GENE_MODULE, KEGG_OXIDATIVE_PHOSPHORYLATION and BILANGES_SERUM_AND_RAPAMYCIN_SENSITIVE_GENES.

Differentially expressed genes from mRNA-seq data were determined using the DESeq2 package of R/Bioconductor, and genes with the mean of reads for all conditions greater than 100 were used for further analysis (*Supplementary file 1e*). Genes with adjusted p-value<0.05 were considered as significantly differentially (*Figures 5–6*). Lists of 5' TOP/TOP-like genes were obtained from *Thoreen et al. (2012)*, whereas the integrated stress response (ISR) gene list was derived from *Palam et al. (2015)*.

Gene ontology analysis was performed using g:Profiler for the TE and DEG datasets (*Reimand et al., 2016*). The START App was employed to generate the volcano plots, in which red dots indicate significantly differentially translated or expressed genes, while green and gray dots indicate non-significantly changed genes (*Nelson et al., 2017*; *Reimand et al., 2016*). Heatmaps were plotted using GENE-E tool from the Broad Institute.

All Ribo-seq data and mRNA-seq described in this paper are available at the Gene Expression Ominibus (http://www.ncbi.nlm.nih.gov/geo/) under accession GSE119365.

## RT-qPCR and western blot analyses

Gene expression was performed by RT-qPCR and western blot analyses as previously described (*Zhang et al., 2018*). Primers are listed in *Supplementary file 3a*. Antibodies used in western blots are described in *Supplementary file 3b*. *Gapdh* was used as the internal control for RT-qPCR. β-actin or eIF2α was used as a loading control for western blots.

## Measurement of protein synthesis

The incorporation of puromycin into nascent polypeptide chains was used as a measure of protein synthesis as described previously (*Schmidt et al., 2009*; *Zhang et al., 2018*). Briefly, erythroid cells from bone marrow (BM) samples were isolated as previously described (*Zhang et al., 2018*). Isolated cells were further subjected to dead-cell removal using the MACS Dead Cell Removal Kit according to the manufacturer's protocol (Miltenyi Biotech). Then, cells were first treated with DMSO or cycloheximide at 100 µg/mL (Sigma-Aldrich, St. Louis, MO), with chloramphenicol at 100 µM (Sigma-Aldrich) or with INK128 at 0.2 µM (LC Laboratories, Woburn, MA) for 15 min at 37°C, followed by the addition of puromycin (Sigma-Aldrich, St. Louis, MO) at 5 µg/mL. Cells were collected after 15, 30, 45 or 60 min for western blot analysis using anti-puromycin antibody to measure protein synthesis.

## Measurements of oxygen consumption rate, mitochondrial mass and mitochondrial DNA

Oxygen consumption rate (OCR) was measured on a Seahorse XFe96 Analyzer with a Seahorse XFe96 FluxPak (Agilent, Santa Clara, CA). Erythroid cells from BM of mice were isolated as previously described (*Zhang et al., 2018*). After recovery in IMDM with 10% FBS and 3 U/mL Epo at 37°C for 1 hr, 300,000 cells/well were seeded onto an XF96 cell culture microplate coated with Cell-tak (Corning, Corning, NY) and subjected to seahorse assay according to the protocol of the manufacturer. Three OCR measurements — basal, ATP-linked and maximal respiration — were performed after sequential injections of 1 µM oligomycin, 1 µM FCCP, and 0.5 µM Antimycin A/Rotenone. Five technical replicas from the same sample were performed for OCR measurement. Mitochondrial mass was determined by Mitotracker through flow cytometry analysis. The mitochondrial DNA (mtDNA) content was measured by qPCR as previously described (*Liu et al., 2017*). Briefly, after DNA isolation from the purified erythroid cells using a DNeasy Blood and Tissue Kit (Qiagen, Germantown), qPCR with specific primers for mtDNA and genomic DNA (gDNA) (*Supplementary file 3a*) was performed to measure the ratio of mtDNA versus gDNA using the ΔΔCt method.

## Enrichment of FL erythroid progenitors for ex vivo culture and differentiation

Erythroid progenitors from E13.5 FLs of Wt + Fe embryos were enriched by magnetic sorting using EasySep Magnet (StemCell Technologies, Vancouver, Canada) as described (*Thom et al., 2014*) (*Figure 7—figure supplement 2*). In brief, total FL cells were mechanically dissociated by pipetting in PBS containing 2% FBS (Atlanta Biologicals, Norcross, GA), 2.5 mM EDTA and 10 mM glucose. Cells were labeled with biotin-conjugated anti-B220, anti-CD3, anti-CD11b, anti-CD11c, anti-GR1, anti-Ter119 (all BioLegend, San Diego, CA) and anti-CD41 (eBioscience, San Diego, CA) antibodies. Lineage negative (Lin⁻Ter119⁻) cells were subjected to a second purification step to obtain Lin⁻Ter119⁻CD71⁻ erythroid progenitors using biotin-conjugated anti-CD71 antibodies (BioLegend, San Diego, CA and BD Biosciences, San Jose, CA).

Purified Lin⁻Ter119⁻CD71⁻ erythroid progenitors were cultured in the expansion medi described by *Thom et al. (2014)* for three days at an initial cell density of $0.5 \times 10^6$ cells per mL. This expansion medium was StemPro-34 medium complemented with 10% supplement, 2 mM L-glutamine, 1% penicillin-streptomycin (P-S), $10^{-4}$ M β-mercaptoethanol, $10^{-6}$ M dexamethasone, 0.5 U/mL Epo, and 100 ng/mL mouse stem cell factor (mSCF). Then, cells were washed and cultured in differentiation medium at an initial cell density of $1 \times 10^6$ cells per mL. The differentiation medium was Iscove-modified Dulbecco medium (IMDM) containing 10% FCS (fetal calf serum, Gemini Bio-Products, West Sacramento, CA), 10% plasma-derived serum (Animal Technologies, Tyler, TX), 2 mM L-glutamine, 1% P-S, $10^{-4}$ M β-mercaptoethanol, and 5 U/mL Epo. Cells were collected for analysis at different time points as indicated in *Figure 7A*.

## Knockdown of *Grb10* expression in FL EBs by shRNAs

The knockdown of *Grb10* expression was performed by using recombinant retroviruses containing an shRNA-expressing murine stem cell retroviral vector, MSCV-pgkGFP-U3-U6P-Bbs, a kind gift from the laboratory of Dr. Harvey F Lodish (MIT). DNA sequences of eight shRNA oligonucleotides (*Supplementary file 3c*) targeting different regions of *Grb10* mRNA were either obtained from the Genetic Perturbation Platform of the Broad Institute or as previously reported (*Doiron et al., 2012*; *Zacharek et al., 2011*; *Park et al., 2013*) and were synthesized by Integrated DNA Technologies. Preparations of the plasmid constructs and recombinant retroviruses were performed as described previously (*Hu et al., 2014*). Lin⁻Ter119⁻CD71⁻ erythroid progenitors enriched from Wt + Fe E13.5 FLs were infected with retroviruses as described previously (*Thom et al., 2014*). Cells were expanded for 72 hr after retroviral infections followed by differentiation for up to 42 hr as indicated.

## Analysis of erythroid differentiation and cell proliferation

Erythroid differentiation was performed by flow cytometry using anti-Ter119 and anti-CD71 antibodies (BioLegend, San Diego, CA) as described previously (*Zhang et al., 2018*) on FACS LSR II (BD Biosciences, San Jose, CA). 4′,6-diamidino-2-phenylindole (DAPI, Roche Diagnostics, Basel, Switzerland) was used to exclude the dead cells. Data were analyzed with FlowJo (Tree Star, Ashland, OR). Erythroid differentiation was also analyzed by cell morphology on cytospin slides stained with May-Grunwald/Giemsa staining (Sigma-Aldrich, St. Louis, MO). Cell proliferation was determination by counting nucleated cells daily using crystal violet stain.

## Statistical analysis

The independent $t$ test (two-tailed) was used to analyze the experimental data. Pearson correlation analysis was performed to determine the correlation coefficient. Data are presented as means ± SEs. *$p < 0.05$ was considered statistically significant. **$p < 0.01$; ***$p < 0.001$.

## Acknowledgements

This paper is dedicated to the memory of Irving M London and his generous and inspiring mentorship. This work was supported by National Institute of Health Grants RO1 DK087984 (to J-JC), and R01 DK103794 and R33 HL120791 (to VGS). We are grateful for the expert help of Dr. Nora Kory from Dr. David Sabatini's lab with Seahorse assays, which were performed at the Whitehead Institute for Biomedical Research, Cambridge, MA.

## Additional information

### Funding

| Funder | Grant reference number | Author |
| --- | --- | --- |
| National Institute of Diabetes and Digestive and Kidney Diseases | RO1 DK087984 | Jane-Jane Chen |
| National Institute of Diabetes and Digestive and Kidney Diseases | R01 DK103794 | Vijay G Sankaran |
| National Heart, Lung, and Blood Institute | R33 HL120791 | Vijay G Sankaran |

The funders had no role in study design, data collection and interpretation, or the decision to submit the work for publication.

### Author contributions

Shuping Zhang, Data curation, Formal analysis, Validation, Investigation, Visualization, Methodology, Writing—original draft; Alejandra Macias-Garcia, Formal analysis, Validation, Investigation, Methodology, Writing—review and editing; Jacob C Ulirsch, Formal analysis, Methodology, Writing—review

and editing; Jason Velazquez, Investigation; Vincent L Butty, Stuart S Levine, Formal analysis; Vijay G Sankaran, Formal analysis, Funding acquisition, Writing—review and editing; Jane-Jane Chen, Conceptualization, Resources, Data curation, Formal analysis, Supervision, Funding acquisition, Investigation, Methodology, Writing—original draft, Project administration

### Author ORCIDs
Shuping Zhang (iD) https://orcid.org/0000-0003-1586-0559
Vijay G Sankaran (iD) https://orcid.org/0000-0003-0044-443X
Jane-Jane Chen (iD) https://orcid.org/0000-0002-4372-6907

### Ethics
Animal experimentation: Mice were maintained at the Massachusetts Institute of Technology (MIT) animal facility, and all experiments were carried out using protocols (#1015-099-18) approved by the Division of Comparative Medicine, MIT.

### Decision letter and Author response
Decision letter https://doi.org/10.7554/eLife.46976.028
Author response https://doi.org/10.7554/eLife.46976.029

## Additional files

### Supplementary files
• Supplementary file 1. List of diifferentially translated mRNAs by HRI and mTORC1. Supplementary File 1a. Quality control and mapping of Ribo-seq and mRNA-seq. Supplementary File 1b. Complete gene list of differentially translated mRNAs. Supplementary File 1c. *Eif2ak1*, *Ppp1r15a* and *Atf4* mRNAs are highly expressed in basophilic erythroblasts. Supplementary File 1d. Higher ribosome occupancy at the AUG of uORF1 in *Atf4* mRNA.Supplementary File 1e. Complete gene list of differentially expressed mRNAs.
DOI: https://doi.org/10.7554/eLife.46976.021

• Supplementary file 2. List of mRNAs that are differentially translated between Hri$^{-/-}$ –Fe EBs and *Wt* –Fe EBs and known mTORC1 targets in the categories of (a) RPs and non-RPs and (b) mitochondrial proteins.
DOI: https://doi.org/10.7554/eLife.46976.022

• Supplementary file 3. List of RNA primers, antibodies and ShRNA oligonucleotides. Supplementary File 3a. DNA sequence of qPCR primers. Supplementary File 3b. List of antibodies used for western blot analyses. Supplementary File 3c. DNA sequences of shRNA oligonucleotides.
DOI: https://doi.org/10.7554/eLife.46976.023

• Transparent reporting form
DOI: https://doi.org/10.7554/eLife.46976.024

### Data availability
All sequencing data have been deposited in GEO under accession code GSE119365.

The following dataset was generated:

| Author(s) | Year | Dataset title | Dataset URL | Database and Identifier |
|---|---|---|---|---|
| Zhang S, Chen J | 2018 | Iron and Heme Coordinate Erythropoiesis through HRI-Mediated Regulation of Protein Translation and Gene Expression | https://www.ncbi.nlm.nih.gov/geo/query/acc.cgi?acc=GSE119365 | NCBI Gene Expression Omnibus, GSE119365 |

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
