## [Decision Letter]

[Editors’ note: a previous version of this study was rejected after peer review, but the authors submitted for reconsideration. The first decision letter after peer review is shown below.]

Thank you for submitting your work entitled "Iron and Heme Coordinate Erythropoiesis through HRI-Mediated Regulation of Protein Translation and Gene Expression" for consideration by *eLife*. Your article has been reviewed by three peer reviewers, one of whom is a member of our Board of Reviewing Editors, and the evaluation has been overseen by a Senior Editor. The reviewers have opted to remain anonymous.

Our decision has been reached after consultation between the reviewers. Based on these discussions and the individual reviews below, we regret to inform you that your work will not be considered further for publication in *eLife*.

While the reviewers found that many aspects of your work were well done, they raise concerns about the degree to which it advances prior studies of similar mechanisms. They were also concerned about some aspects of the mechanisms, and the strength of conclusions that could be drawn based on loss-of-function studies with a single shRNA and experiments performed in culture. We believe your paper would be more appropriate for a more specialized journal.

Reviewer #1:

In this manuscript, Chen and colleagues characterized how iron, heme and the heme-regulated kinase HRI coordinate to regulate erythropoiesis under diet-induced iron deficiency (ID). By unbiased and comparative analysis of ribosome profiling and mRNA-seq datasets in mouse fetal liver basophilic erythroblasts (EBs) with or without ID, they validated the known regulator ATF4 as the most differentially translated mRNA in HRI and iron deficiencies. They also identified additional new pathways such as ribosomal protein mRNA translation. The authors then focused on HRI-ATF4 and described ATF4 as the major transcription factor under HRI translational control likely through uORF regulation. They also showed that heme-HRI-ATF4 pathway, but not iron-regulated pathways such as IRE-containing mRNAs, mediate the major translational and transcriptional changes in response to iron deficiency. Finally, they identified *Grb10* as one of the ATF4 target genes that may function to inhibit Epo-mTORC1 in ID.

Overall, this study presents a set of interesting findings related to the role of heme-HRI-ATF4 pathway in erythropoiesis and iron deficiency anemia. These results not only validated ATF4 as a major factor in coordinating protein translation and gene expression during ID, but also identified regulators such as *Grb10* in regulating erythropoiesis for the adaptation to iron deficiency anemia. In this regard, this manuscript expanded our understanding of the underlying basis for iron deficiency or ineffective erythropoiesis. The ribosome and mRNA profiling studies were appropriately designed and carefully analyzed. There are several important questions remained to be addressed related to the mechanistic details how ATF4 regulates *Grb10* and how *Grb10* regulates Epo-mTORC1 in the context of erythropoiesis. Moreover, the functional roles of these pathways in more physiologically relevant conditions need to be established. If the remaining questions can be adequately addressed, this study may provide important insights into heme-HRI-mediated pathways in iron deficiency and ineffective erythropoiesis, and be potentially impactful.

1) It is difficult to follow and somewhat confusing on the various comparisons of the ribosome profiling and mRNA-seq in WT+Fe, HRI^-/-^+Fe, WT-Fe and HRI^-/-^-Fe (Figures 2-4, Figure 2—figure supplement 1). It would be helpful if the authors could summarize the findings in a diagram to list the most differentially regulated pathways and/or genes in the pairwise comparison. This diagram can be combined in Figure 7 or as a separate figure.

2) When comparing differentially expressed mRNAs in HRI and iron deficiencies, the authors identified *Grb10* as one of the most upregulated ATF4 target genes in WT relative to HRI^-/-^ basophilic erythroblasts. To establish the functional role of *Grb10* in erythropoiesis, a single shRNA (G3) was used to knockdown *Grb10* expression in ex vivo cultured erythroid cells. The authors measured phosphorylated AKTS473, which is slightly increased, as the evidence for the role of *Grb10* as the negative regulator of mTORC1 signaling. It is important to characterize in greater details how *Grb10* regulate mTORC1 or other pathways in erythropoiesis. The authors cited previous studies (Plasschaet and Bartolomer, 2015, and Yan et al., 2016) as evidence that *Grb10* is part of a feedback mechanism to inhibit mTORC1, however it is important to determine whether similar or different mechanism may be utilized in the context of erythropoiesis. How does ATF4 regulate *Grb10* transcription (or translation) in erythroid cells? Moreover, additional shRNAs should be screened to validate the original findings. Since this part of the study could provide potentially new insights compared to the already established HRI-ATF4 pathway in erythropoiesis, more in-depth mechanistic studies are required.

3) Since all the functional studies were performed using ex vivo cultured erythroid cells with defined medium and cytokines, the functional importance of the identified pathway (HRI-ATF4-*Grb10*) should be assessed in more physiologically relevant settings in vivo, such as in the already established *Grb10* knockout mouse models or in the transplant setting.

Reviewer #2:

The manuscript by Zhang et al. describes an analysis of global mechanisms regulating protein translation by iron, heme and HRI in primary erythroblasts. HRI has been studied for many years by Dr. Chen, and it represents a component of cellular machinery that senses alterations in heme. Other cellular mechanisms of high importance also exist, including the IRE-IRP system, as well as transcriptional mechanisms involving the heme-regulated repressor BACH1. A major conclusion of the RNA-seq and Ribo-seq analyses is that iron deficiency most highly activates the integrated stress response and ATF4-mediated gene expression. While the studies appear to have been rigorously conducted, and the overall problem is significant, this major conclusion is not surprising based on prior published work. The global analyses did not transform or extend existing knowledge in a major way, which represents a significant limitation. Recommendations are presented below to build upon the work to elevate novelty and broad interest.

1) Subsection “Upregulation of in vivo translation of ISR mRNAs in *Wt* EBs compared to *Hri*^-/-^ EBs”, last paragraph – The authors conclude that diet-induced iron deficiency does not significantly affect gene expression through IREs/IPRs in primary erythroblasts. I do not see how the global measurements and correlations support this rather strong conclusion. At the very least, this needs to be fleshed out more effectively.

2) "ATF4 target gene GRB10". GRB10 has been published by Mehta et al., 2017, to be a direct GATA2 target gene in primary mouse erythroid precursor cells. This may influence thinking regarding the role of GRB10 in mechanisms of interest to the authors.

3) Figure 6 – as noted above, the finding that Integrated Stress Response and ATF4 are important players does not transform existing knowledge. A limited loss of function analysis was performed with retrovirally-expressed shRNAs to evaluate GRB10 function. Only one shRNA was effective in reducing GRB10 levels in the differentiation culture. It will be important not to rely on results obtained with a single shRNA, especially given the recent findings by Traxler et al. (2017) Blood 131, 2733-2736 that shRNAs can often "non-specifically" inhibit erythroid terminal differentiation independent of their specific target – and the GRB10 knockdown phenotype with the single shRNA is inhibition of terminal differentiation.

4) Given the datasets generated, and the fact that GRB10 has been published as a GATA2 target gene in erythroid precursor cells and as a regulator of cKit signaling in HSCs (Yan et al., 2016), can additional loss of function studies be conducted to yield discoveries that considerably extend the existing work – for example to implicate proteins and/or pathways that have not been described as being important in the erythroid (or hematopoietic) system?

5) The Discussion was quite extensive, but does not seem to effectively integrate existing knowledge of translational and transcriptional mechanisms mediating iron- and heme-dependent cell regulation. Integration is important, since we know about various components, but very little is known about integration.

Reviewer #3:

The authors present extensive data concerning the differential translational of mRNA during erythtroid differentiation of mouse fetal liver cells in vitro. Although the results are interesting, they add little to the current knowledge vis-à-vis the main proteins involved in erythroid differentiation. The text of the manuscript is flooded with an overwhelming amount of data that is difficult to read, analyze and summarize. In other words, in the middle of so much data, it is hard pinpoint the main take-home message of the current article.

The authors claim that they have noticed only small changes in the translation of IRE-containing mRNAs when comparing normal and iron deficient conditions. The authors specifically mentioned that there are no differences in the translation of ALAS2 and TfR1 mRNA when compared normal and iron deficient conditions. The authors should extend the discussion regarding these findings, since they are in conflict with what has been shown in the literature in the past 30 years.

Moreover, it is necessary to better explain how the iron deficient condition was attained. According to the Materials and methods section, mice were kept on an iron deficient diet for an unspecified period, following which it was observed that the embryos obtained from these mice were "pale". The visual appearance of the embryos, although not unimportant, cannot be used as measure of the iron status of the embryos. A simple measurement of hemoglobin levels in the fetal livers would have directly answered that the cells were lacking iron.

Furthermore, the authors mentioned several papers published in 2015 and 2016 claiming that the protein NCOA4 is important for erythroid differentiation, since it is required for iron mobilization from ferritin. It is important to stress that these are questionable findings since the NCO4A knockout mice show hypochromic microcytic anemia only when exposed to an iron deficient diet; this is not at all surprising. In this context, unfortunately, the authors disregard extensive literature supporting the view that ferritin does not supply iron for heme and hemoglobin synthesis. These references are neglected by the authors when discussing the rate of translation of ferritin in their in vitro system.

[Editors’ note: what now follows is the decision letter after the authors submitted for further consideration.]

Thank you for resubmitting your work entitled "HRI coordinates translation necessary for protein homeostasis and mitochondrial function in erythropoiesis" for further consideration at *eLife*. Your revised article has been favorably evaluated by David Ron (Senior Editor), a Reviewing Editor, and 2 reviewers.

The manuscript has been improved and represents a significant extension of prior work on HRI function in erythropoiesis. As you will see from their comments (attached below) the reviewers find the new results of interest, but there are some remaining issues that need to be addressed before acceptance, as outlined below:

- As reviewer #1 points out, questions about mTORC1 activity in *Hri*^-/-^ -Fe vs *Hri*^-/-^ +Fe erythroid cells will need to be assessed.

- As reviewer #1 points out, questions about the strength of several statements on the role of HRI and mTORC1 in regulating translation of mitochondrial genes (point #2) and difference in mtDNA and mitochondrial mass (point #3) will need to be addressed with further clarifications.

- As reviewer #2 points out, the statement on GRB10 as a new regulator of erythropoiesis will need to be revised and include the relevant citation(s).

- All other reviewer concerns pertaining to strengthening existing data, discussion, clarifications and textual changes should also be addressed.

We therefore invite you to take these points into account when revising the manuscript. We look forward to receiving your revised manuscript and ask that you return it within four weeks. Please feel free to contact me if you require additional time or would like to discuss any of the reviewer comments further.

Reviewer #1:

In the revised manuscript, the authors removed some of the original results (i.e. IRE/IRPs and NCOA4) and shifted focus to an interesting new link between HRI-ATF4 and the translation of cytosolic and mitochondrial ribosomal proteins in the context of iron-deficiency (ID)-induced ineffective erythropoiesis. They also re-organized the presentation of the Ribo-seq and RNA-seq comparison in the first part of the manuscript. These efforts significantly improved both the strength of new conclusions and the clarity of the main messages. In particular, the new results support an important role of the heme-induced kinase HRI in coordinating integrated stress response (ISR), translation of cytosolic and mitochondrial proteins, and Epo-mTORC1 signaling during normal and iron-deficiency-induced ineffective erythropoiesis. These new findings significantly extend previous results, and provide new insights into the coordinated regulation of protein translation, mitochondria function, and ISR during normal and ineffective erythropoiesis. They also provide additional evidence for the role of GRB10 as a negative feedback regulator of Epo-mTORC1 and an ATF4 target gene during ID, and discuss other mechanisms (e.g. transcriptional regulation by ATF4 and/or GATA2) that may contribute to GRB10 regulation during erythropoiesis.

Overall, the authors provide a nice resource of transcriptional (mRNA-seq) and translational (Ribo-seq) changes in response to HRI loss during normal and ID-induced ineffective erythropoiesis, which will be of great interest to the erythropoiesis and a broader community. The new findings related to HRI-coordinated control of protein translation, ISR and mitochondrial function advance our current understanding of mechanisms that regulate normal and ineffective erythropoiesis.

There are several remaining questions and suggestions for area of improvement, which if addressed, will further enhance the strength of the conclusions and the potential impact of this study.

1) Is mTORC1 activity increased in *Hri*^-/-^ -Fe erythroid cells compared to *Hri*^-/-^ +Fe? From GSEA analysis of differentially translated mRNAs between *Hri*^-/-^ -Fe and *Wt* -Fe (Figure 3C), it seems that mTORC1 signaling is significantly elevated, consistent with the role of HRI in negative regulation of mTORC1. However, it is unclear from the current analysis whether any changes in mTORC1 activity between *Hri*^-/-^ -Fe and *Hri*^-/-^ +Fe erythroid cells, or between *Wt* -Fe and *Wt* +Fe cells. Furthermore, the authors should consider measuring mTORC1 activity directly by the level of phosphorylated 4EBP1 and/or other downstream targets as in Figure 2E.

2) In the Discussion section (second paragraph), the authors discussed that the mitochondrial genes regulated by HRI are different from the targets of mTORC1, and suggested that HRI may regulate translation of mitochondrial genes independent of mTORC1. This statement is a bit confusing. The evidence supporting this statement is based on the lack of overlapped targets for mitochondria related proteins in *Hri*^-/-^ -Fe vs mTORC1 (Figure 3D). Since HRI may negatively regulate mTORC1 through GRB10, one would expect to see some overlapped targets if HRI-GRB10-mTORC1 mechanism plays a significant role in erythropoiesis. Furthermore, HRI loss may also impact transcriptional programs and/or indirectly affect mitochondrial proteins independent of protein translation. The authors should revise the statement and/or provide additional clarification.

3) To determine the effect of *Hri* loss on mitochondria, the authors used mitotracker to estimate mitochondrial mass. It is important to note that mitotracker signal may also be affected by mitochondrial activity. In Figure 4—figure supplement 1, it seems that the mtDNA content is modestly increased in *Hri*^-/-^ -Fe cells. Is the difference statistically significant? If the difference is significant, the results would suggest that *Hri* loss affect both mitochondrial biogenesis (mtDNA) and mitochondrial mass (or activity).

4) Some parts of the manuscript are still difficult to follow and confusing with various comparisons. The authors are advised to further improve the clarity and presentation of the manuscript by removing and/or simplifying non-essential texts, and focus more on main messages. The Discussion is extensive and could be further compressed.

5) Several typos and grammar errors are noted: Figure 3D legend "Venn" diagram, subsection “Increased cytoplasmic and mitochondrial protein synthesis but reduced mitochondrial respiratory activity in *Hri*^-/-^-Fe EBs”, first paragraph etc.

Reviewer #2:

The manuscript is improved and represents a significant extension of prior work on HRI function in the context of erythropoiesis. Dr. Chen has conducted foundational work on HRI for many years, and this new work provides a new dimension that largely could not have been predicted from known mechanistic or biological concepts. As HRI was implicated recently by Gerd Blobel as a regulator of fetal hemoglobin production, there is renewed interest in the function of this important protein.

One minor, but significant, comment remains, which was not addressed in the revision.

The authors indicate in the cover letter and have related text in the manuscript that their study demonstrates: "These additional results strengthened our finding GRB10 as a previously unappreciated regulator of fetal liver erythropoiesis".

However, GRB10 was already published in Mehta et al., 2017, to be a direct GATA2 target gene in primary fetal liver erythroid precursor cells. Based on this work and known GRB10 biochemical activities, GRB10 is not a "previously unappreciated regulator of fetal liver erythropoiesis". This work should be cited in the context of what was already known about GRB10 in erythropoiesis.

---

## [Author Response]

[Editors’ note: the author responses to the first round of peer review follow.]

While the reviewers found that many aspects of your work were well done, they raise concerns about the degree to which it advances prior studies of similar mechanisms. They were also concerned about some aspects of the mechanisms, and the strength of conclusions that could be drawn based on loss-of-function studies with a single shRNA and experiments performed in culture. We believe your paper would be more appropriate for a more specialized journal.

The new manuscript is revised extensively and emphasizes on the translational regulation of HRI in iron deficiency (ID) to maintain protein homeostasis and mitochondrial function during erythropoiesis. Our global study supports an integrated model of heme and HRI regulated translation in developing erythroblasts for the adaptation to systemic ID (Figure 8). Activation of HRI in ID coordinates three distinct pathways of translation. First and foremost, inhibition of protein synthesis to avoid accumulation of unfolded proteins and maintain protein homeostasis. Second, induction of ATF4 mRNA translation to increase gene expression for mitochondrial unfolded protein response, redox homeostasis and metabolic reprogramming in order to maintain mitochondrial respiration and erythroid differentiation. Third, repression of Epo- mTORC1 signaling to inhibit protein synthesis and cell proliferation.

We provide new data on the novel findings of heme and HRI mediated translation of mRNAs of both cytosolic and mitochondrial ribosomal proteins in ID concomitant with inhibition of protein synthesis in both cellular compartments (Figure 3). The absence of HRI in ID elicits a prominent cytoplasmic unfolded protein response (UPR) in attempt to reduce proteotoxicity of excessive unfolded proteins produced as evidenced by the comprehensive induction of several families of cytoplasmic chaperones (Figure 5A-B).

Importantly, neither endoplasmic reticulum (ER) nor mitochondrial chaperones were induced. The lack of ER UPR is consistent with our earlier observation that erythroblasts do not display significant ER stress and that HRI is activated by cytoplasmic, but not ER stress (Lu L, Han AP, Chen JJ. Mol Cell Biol. 2001;21(23):7971-7980).

Integrated stress response (ISR) of eIF2a kinase has been shown to be activated in mitochondrial stress and mediates mitochondrial UPR. To date, the eIF2a kinase responsible for mitochondrial UPR remains unknown. The lack of mitochondrial UPR in HRI deficiency shown in our present study supports the notion that HRI is necessary for the induction of mitochondrial UPR in ID.

We also discover that Hri -/- primary erythroid cells have impairment of mitochondrial respiration (Figure 4). Importantly, ATF4 target genes are most highly activated during ID including serine-glycine biosynthesis, one carbon metabolism and redox homeostasis, for the maintenance of mitochondrial function and enable erythroid differentiation. We now have two shRNAs for Grb10 knockdown study, and showed that the additional shRNA_G7 worked similarly to shRNA_G3 in increasing cell proliferation and inhibiting erythroid

differentiation (Figure 7). These additional results strengthened our finding of GRB10 as a previously unappreciated regulator of fetal liver erythropoiesis connecting between HRI and erythropoietin signaling during ID.

Reviewer #1:[…] 1) It is difficult to follow and somewhat confusing on the various comparisons of the ribosome profiling and mRNA-seq in WT+Fe, HRI^-/-^+Fe, WT-Fe and HRI^-/-^-Fe (Figures 2-4, Figure 2—figure supplement 1). It would be helpful if the authors could summarize the findings in a diagram to list the most differentially regulated pathways and/or genes in the pairwise comparison. This diagram can be combined in Figure 7 or as a separate figure.

We thank the reviewer for the suggestion, and have expand the diagram in the model (Figure 8 in the updated version) summarizes our findings of ribosome profiling and mRNA-seq. This figure is referred in the Results section for the clarity of data presented.

2) When comparing differentially expressed mRNAs in HRI and iron deficiencies, the authors identified Grb10 as one of the most upregulated ATF4 target genes in WT relative to HRI^-/-^ basophilic erythroblasts. To establish the functional role of Grb10 in erythropoiesis, a single shRNA (G3) was used to knockdown Grb10 expression in ex vivo cultured erythroid cells. The authors measured phosphorylated AKTS473, which is slightly increased, as the evidence for the role of Grb10 as the negative regulator of mTORC1 signaling. It is important to characterize in greater details how Grb10 regulate mTORC1 or other pathways in erythropoiesis. The authors cited previous studies (Plasschaet and Bartolomer, 2015, and Yan et al., 2016) as evidence that Grb10 is part of a feedback mechanism to inhibit mTORC1, however it is important to determine whether similar or different mechanism may be utilized in the context of erythropoiesis. How does ATF4 regulate Grb10 transcription (or translation) in erythroid cells? Moreover, additional shRNAs should be screened to validate the original findings. Since this part of the study could provide potentially new insights compared to the already established HRI-ATF4 pathway in erythropoiesis, more in-depth mechanistic studies are required.

We now have an additional shRNA_G7, which maintained knockdown efficiency at differentiation stage. Similar results were obtained by G7 and G3 shRNA in increasing cell proliferation and inhibition of erythroid differentiation (Figure 7). We found that protein level of cyclin D3, which is a known target of mTORC1 signaling, was increased upon Grb10 knockdown, supporting the role of Grb10 in repressing mTORC1 pathway. These are incorporated into the updated manuscript.

Our ribosome profiling and mRNA-seq results showed that *Grb10* was regulated at transcription and not translation by ATF4. This is consistent with a recent study demonstrating that *Grb10* expression is activated upon acute ER stress through ATF4 (Luo et al. 2018, J. Mol. Endocrinol. 60, 285).

We agree with the reviewer that it will be important to determine the molecular mechanism by which ATF4 regulates *Grb10* expression as well as by which GRB10 regulates mTORC1 signaling. We feel these are outside the scope of the present study. In the updated manuscript, we focused on the global assessment and impact by HRI-mediated translation on ribosome synthesis and mitochondrial function.

3) Since all the functional studies were performed using ex vivo cultured erythroid cells with defined medium and cytokines, the functional importance of the identified pathway (HRI-ATF4-Grb10) should be assessed in more physiologically relevant settings in vivo, such as in the already established Grb10 knockout mouse models or in the transplant setting.

We agree with the reviewer, but feel it is also outside the scope of this study in the updated manuscript as described above.

Reviewer #2:[…] 1) Subsection “Upregulation of in vivo translation of ISR mRNAs in Wt EBs compared to Hri^-/-^ EBs”, last paragraph – The authors conclude that diet-induced iron deficiency does not significantly affect gene expression through IREs/IPRs in primary erythroblasts. I do not see how the global measurements and correlations support this rather strong conclusion. At the very least, this needs to be fleshed out more effectively.

We have removed IRE/IRP results from the updated manuscript. Perhaps, much more severe ID is necessary to detect IRE/IRP mediated translation in erythroid cells. The major point to be conveyed in our manuscript is that HRI plays a major role in gene expression of EBs during ID. Globally, translation of ribosomal proteins, both cytosolic and mitochondrial, is the most highly differentially regulated between *Wt* and *Hri^-/-^*EBs in ID. At the mRNA level, ATF4-ISR target genes were most highly activated in *Wt-Fe* EBs compared to *Hri^-/-^*-Fe EBs.

2) "ATF4 target gene GRB10". GRB10 has been published by Mehta et al., 2017, to be a direct GATA2 target gene in primary mouse erythroid precursor cells. This may influence thinking regarding the role of GRB10 in mechanisms of interest to the authors.

We thank reviewer for bring our attention to this line of thoughts. In the study of Mehta *et al. Grb10* was suggested to participate in the transition of BFU-E to CFU-E, which are earlier in erythroid differentiation than the EBs that we studied here. We will keep this in mind in our future study of erythropoiesis at early stage of erythropoiesis.

Since last submission of our manuscript, there was a recent report showing the ATF4-dependent *Grb10* induction upon ER stress as responded above to reviewer 1. Our results also show decreased *Grb10* expression during ex vivodifferentiation in *Hri^-/-^, eAA* and *Atf4^-/^*^-^ fetal liver progenitors compared to *Wt*. Thus, there is a strong experimental support for the role of *Grb10* in terminal erythropoiesis, where GATA2 expression is low.

3) Figure 6 – As noted above, the finding that Integrated Stress Response and ATF4 are important players does not transform existing knowledge. A limited loss of function analysis was performed with retrovirally-expressed shRNAs to evaluate GRB10 function. Only one shRNA was effective in reducing GRB10 levels in the differentiation culture. It will be important not to rely on results obtained with a single shRNA, especially given the recent findings by Traxler et al. (2017) Blood 131, 2733-2736 that shRNAs can often "non-specifically" inhibit erythroid terminal differentiation independent of their specific target – and the GRB10 knockdown phenotype with the single shRNA is inhibition of terminal differentiation.

We agree with the reviewer and have an additional shRNA_G7, which worked similarly to G3.

4) Given the datasets generated, and the fact that GRB10 has been published as a GATA2 target gene in erythroid precursor cells and as a regulator of cKit signaling in HSCs (Yan et al., 2016), can additional loss of function studies be conducted to yield discoveries that considerably extend the existing work – for example to implicate proteins and/or pathways that have not been described as being important in the erythroid (or hematopoietic) system?

We thank the reviewer for the interesting suggestion. We believe this is outside the scope of this study presently as addressed above in response to the comments of reviewer 1, point 2. In updated and extensively revised manuscript, we focused on the global assessment and impact by HRI-mediated translation on ribosome synthesis and mitochondrial function.

5) The Discussion was quite extensive, but does not seem to effectively integrate existing knowledge of translational and transcriptional mechanisms mediating iron- and heme-dependent cell regulation. Integration is important, since we know about various components, but very little is known about integration.

We have rewritten the Discussion and presented adistinct model of heme and HRI-regulated translational control that involves several separate and interacting pathways for the adaptation to systemic ID (Figure 8). Activation of HRI in ID elicits three distinct pathways of translation as described in the Discussion of the revised manuscript.

Reviewer #3:The authors present extensive data concerning the differential translational of mRNA during erythtroid differentiation of mouse fetal liver cells in vitro. Although the results are interesting, they add little to the current knowledge vis-à-vis the main proteins involved in erythroid differentiation. The text of the manuscript is flooded with an overwhelming amount of data that is difficult to read, analyze and summarize. In other words, in the middle of so much data, it is hard pinpoint the main take-home message of the current article.

We appreciate the helpful comments and have revised the manuscript extensively, not only in writing, but also in the presentation in figures.

The authors claim that they have noticed only small changes in the translation of IRE-containing mRNAs when comparing normal and iron deficient conditions. The authors specifically mentioned that there are no differences in the translation of ALAS2 and TfR1 mRNA when compared normal and iron deficient conditions. The authors should extend the discussion regarding these findings, since they are in conflict with what has been shown in the literature in the past 30 years.

This is addressed above in response to comments of Reviewer 2, point 1.

Moreover, it is necessary to better explain how the iron deficient condition was attained. According to the Materials and methods section, mice were kept on an iron deficient diet for an unspecified period, following which it was observed that the embryos obtained from these mice were "pale". The visual appearance of the embryos, although not unimportant, cannot be used as measure of the iron status of the embryos. A simple measurement of hemoglobin levels in the fetal livers would have directly answered that the cells were lacking iron.

We have added the details of inducing ID in developing embryos in the Materials and methods. We have shown in our earlier publication that under our iron deficient conditions, E14.5 *Hri^-/-^*embryos were anemic with prolong primitive erythrocytes in the embryonic blood.

Furthermore, inclusions were observed in both primitive and definitive erythrocytes (Liu et al., 2008). *Hri^-/-^*embryos in ID died at E17.5.

Furthermore, the authors mentioned several papers published in 2015 and 2016 claiming that the protein NCOA4 is important for erythroid differentiation, since it is required for iron mobilization from ferritin. It is important to stress that these are questionable findings since the NCO4A knockout mice show hypochromic microcytic anemia only when exposed to an iron deficient diet; this is not at all surprising. In this context, unfortunately, the authors disregard extensive literature supporting the view that ferritin does not supply iron for heme and hemoglobin synthesis. These references are neglected by the authors when discussing the rate of translation of ferritin in their in vitro system.

We have removed the discussion about IRE/IRP and NCOA4 in the updated manuscript and focus on HRI pathway.

[Editors' note: the author responses to the re-review follow.]

Reviewer #1:[…] 1) Is mTORC1 activity increased in Hri^-/-^ -Fe erythroid cells compared to Hri^-/-^ +Fe? From GSEA analysis of differentially translated mRNAs between Hri^-/-^ -Fe and Wt -Fe (Figure 3C), it seems that mTORC1 signaling is significantly elevated, consistent with the role of HRI in negative regulation of mTORC1. However, it is unclear from the current analysis whether any changes in mTORC1 activity between Hri^-/-^ -Fe and Hri^-/-^ +Fe erythroid cells, or between Wt -Fe and Wt +Fe cells. Furthermore, the authors should consider measuring mTORC1 activity directly by the level of phosphorylated 4EBP1 and/or other downstream targets as in Figure 2E.

mTORC1 activity in *Hri*^+/+^ and *Hri*^-/-^ erythroid precursors in +Fe and -Fe conditions were published last year in Zhang et al., 2018 (https://doi.org/10.1182/blood-2017-08-799908, Supplementary Figure 3A). mTORC1 activities as measured by pS6 and p4EBP1 were increased in primary *Hri*^-/-^-Fe erythroid cells from blood, bone marrow and spleen as compared to *Hri*^-/-^+Fe cells. This is consistent with the mild erythroid phenotypes of *Hri*^-/-^ mice in +Fe as compared to -Fe conditions.

TE analysis of ribosome profiling data in the present manuscript also showed increased TE for mRNAs of mTORC1 targets such as cytosolic ribosomal proteins, *Eef1a1, Eef2*, and *Pabpc1* in *Hri*^-/-^-Fe cells as compared to *Hri*^-/-^+Fe (Supplementary file 1B). These results support the increased mTORC1 activity in *Hri*^-/-^-Fe cells as compared to *Hri*^-/-^+Fe cells. In contrast, TE for mTORC1 target mRNAs was not increased in *Wt*-Fe cells as compared to *Wt*+Fe cells (Supplementary file 1B).

2) In the Discussion section (second paragraph), the authors discussed that the mitochondrial genes regulated by HRI are different from the targets of mTORC1, and suggested that HRI may regulate translation of mitochondrial genes independent of mTORC1. This statement is a bit confusing. The evidence supporting this statement is based on the lack of overlapped targets for mitochondria related proteins in Hri^-/-^ -Fe vs mTORC1 (Figure 3D). Since HRI may negatively regulate mTORC1 through GRB10, one would expect to see some overlapped targets if HRI-GRB10-mTORC1 mechanism plays a significant role in erythropoiesis. Furthermore, HRI loss may also impact transcriptional programs and/or indirectly affect mitochondrial proteins independent of protein translation. The authors should revise the statement and/or provide additional clarification.

We thank reviewer for pointing this out. We have employed INK128, a mTORC1 inhibitor to discern the contribution of HRI directly in the regulation of mitochondrial protein synthesis. As compared to *Wt*-Fe cells, *Hri*^-/-^-Fe cells has increased mitochondrial protein synthesis, which was inhibited by INK128 to about 50% of the control (Figure 3—figure supplement 3A). However, mTORC1 activities as measured by pS6 and p4EBP1, were completely inhibited by INK128 treatment (Figure 3—figure supplement 3B). These results demonstrate that HRI also inhibits directly mitochondrial protein synthesis in addition to repress mTORC1 activity during iron deficiency. Thus, both HRI-eIF2αP and mTORC1 contribute to regulation of mitochondrial protein synthesis in erythroid cells during iron deficiency (subsection “Increased cytoplasmic and mitochondrial protein synthesis but reduced mitochondrial respiratory activity in *Hri*^-/-^ -Fe EBs”, first paragraph and Discussion).

We have now modified the statement in Discussion to “both HRI-eIF2αP and mTORC1 pathways contribute to increased mitochondrial protein synthesis in *Hri*^-/-^-Fe EBs”.

3) To determine the effect of Hri loss on mitochondria, the authors used mitotracker to estimate mitochondrial mass. It is important to note that mitotracker signal may also be affected by mitochondrial activity. In Figure 4—figure supplement 1, it seems that the mtDNA content is modestly increased in Hri^-/-^ -Fe cells. Is the difference statistically significant? If the difference is significant, the results would suggest that Hri loss affect both mitochondrial biogenesis (mtDNA) and mitochondrial mass (or activity).

The difference in mtDNA content was not statistically significant as stated in the manuscript.

4) Some parts of the manuscript are still difficult to follow and confusing with various comparisons. The authors are advised to further improve the clarity and presentation of the manuscript by removing and/or simplifying non-essential texts, and focus more on main messages.

We have removed the comparison of ISR and Epo target genes (Figure 5C and 5E) from the main text to supplementary materials (Figure 6—figure supplement 1A and D) to simplify the main text and to improve clarity.

The Discussion is extensive and could be further compressed.

We have compressed the Discussion in the revised manuscript.

5) Several typos and grammar errors are noted: Figure 3D legend "Venn" diagram, subsection “Increased cytoplasmic and mitochondrial protein synthesis but reduced mitochondrial respiratory activity in Hri^-/-^-Fe EBs”, first paragraph etc.

We have corrected the typo and grammar errors.

Reviewer #2:[…] One minor, but significant, comment remains, which was not addressed in the revision.The authors indicate in the cover letter and have related text in the manuscript that their study demonstrates: "These additional results strengthened our finding GRB10 as a previously unappreciated regulator of fetal liver erythropoiesis".However, GRB10 was already published in Mehta et al., 2017 to be a direct GATA2 target gene in primary fetal liver erythroid precursor cells. Based on this work and known GRB10 biochemical activities, GRB10 is not a "previously unappreciated regulator of fetal liver erythropoiesis". This work should be cited in the context of what was already known about GRB10 in erythropoiesis.

We thank the reviewer for the comments. We have modified the Abstract to state that GRB10 as a previously unappreciated regulator of the terminal erythroid differentiation in fetal liver erythropoiesis. We have also included the citation and the role of *Grb10* in the Discussion. *Grb10* has also been shown to be a GATA2 target gene and play a role in the transition from BFU-E to CFU-E (Mehta et al., 2017). Our results support an additional role of ATF4 induced *Grb10* expression in terminal erythropoiesis in which GATA2 expression is low.